

# Were early pterosaurs inept terrestrial locomotors?

Mark P. Witton

School of Earth and Environmental Sciences, University of Portsmouth, Portsmouth, UK

## ABSTRACT

Pterodactyloid pterosaurs are widely interpreted as terrestrially competent, erect-limbed quadrupeds, but the terrestrial capabilities of non-pterodactyloids are largely thought to have been poor. This is commonly justified by the absence of a non-pterodactyloid footprint record, suggestions that the expansive uropatagia common to early pterosaurs would restrict hindlimb motion in walking or running, and the presence of sprawling forelimbs in some species. Here, these arguments are re-visited and mostly found problematic. Restriction of limb mobility is not a problem faced by extant animals with extensive fight membranes, including species which routinely utilise terrestrial locomotion. The absence of non-pterodactyloid footprints is not necessarily tied to functional or biomechanical constraints. As with other fully terrestrial clades with poor ichnological records, biases in behaviour, preservation, sampling and interpretation likely contribute to the deficit of early pterosaur ichnites. Suggestions that non-pterodactyloids have slender, mechanically weak limbs are demonstrably countered by the proportionally long and robust limbs of many Triassic and Jurassic species. Novel assessments of pterosaur forelimb anatomies conflict with notions that all non-pterodactyloids were obligated to sprawling forelimb postures. Sprawling forelimbs seem appropriate for species with ventrally-restricted glenoid articulations (seemingly occurring in rhamphorhynchines and campylognathoidids). However, some early pterosaurs, such as *Dimorphodon macronyx* and wukongopterids, have glenoid arthrologies which are not ventrally restricted, and their distal humeri resemble those of pterodactyloids. It seems fully erect forelimb stances were possible in these pterosaurs, and may be probable given proposed correlation between pterodactyloid-like distal humeral morphology and forces incurred through erect forelimb postures. Further indications of terrestrial habits include antungual sesamoids, which occur in the manus and pes anatomy of many early pterosaur species, and only occur elsewhere in terrestrial reptiles, possibly developing through frequent interactions of large claws with firm substrates. It is argued that characteristics possibly associated with terrestriality are deeply nested within Pterosauria and not restricted to Pterodactyloidea as previously thought, and that pterodactyloid-like levels of terrestrial competency may have been possible in at least some early pterosaurs.

Corresponding author
Mark P. Witton,
mark.witton@port.ac.uk

## INTRODUCTION

The terrestrial competency of pterosaurs was keenly debated during the 1980s and 1990s, when the utility of bipedal and quadrupedal stances, orientation and posture of the extremities, as well as overall terrestriality were discussed at length (Fig. 1, *Padian, 1983a*; *Padian, 1983b*; *Padian & Olsen, 1984*; *Wellnhofer, 1988*; *Unwin, 1988*; *Unwin, 1989*; *Unwin, 1996a*; *Unwin, 1999*; *Lockley et al., 1995*; *Bennett, 1997a*; *Clark et al., 1998*; see *Witton, 2013* for a recent overview). The current consensus emerged when *Pteraichnus* trackways, first identified by *Stokes (1957)* as pterosaurian, but argued to be of crocodylomorph origin by *Padian & Olsen (1984)* and *Unwin (1989)*, were convincingly demonstrated as belonging to pterodactyloid pterosaurs (*Lockley et al., 1995*; *Bennett, 1997a*; *Unwin, 1996a*; *Unwin, 1999*; also see *Kubo, 2008*). This reappraisal started the construction of a compelling case for pterodactyloids as terrestrially competent quadrupeds with plantigrade feet and parasagittal gaits, a hypothesis now strengthened by numerous trackway discoveries (e.g., *Mazin et al., 1995*; *Mazin et al., 2003*; *Lockley & Wright, 2003*; *Hwang et al., 2002*; *Padian, 2003*; *Lockley, Harris & Mitchell, 2008*) as well as functional analyses of pterosaur anatomy (e.g., *Bennett, 1997a*; *Clark et al., 1998*; *Sangster, 2003*; *Wilkinson, 2008*; *Witton & Naish, 2008*; *Fujiwara & Hutchinson, 2012*; *Costa, Rocha-Barbosa & Kellner, 2014*; *Hyder, Witton & Martill, 2014*).

Although it seems that the basic tenets of pterodactyloid terrestrial locomotion are understood, the same cannot be said for non-pterodactyloids. Research into the terrestrial capacity of early pterosaurs is entirely based on interpretations of their functional anatomy because their trackways remain elusive (*Unwin, 2005*; *Lockley, Harris & Mitchell, 2008*; *Whyte & Romano, 2014*). Such considerations are relatively few in number and have reached varying conclusions, either arguing for non-pterodactyloids as terrestrially competent, digitigrade bird-like bipeds which could not easily reach the substrate with their forelimbs (Fig. 1A; *Padian, 1983a*; *Padian, 1983b*; *Padian, 1985*; *Padian, 2003*; *Padian, 2008a*; *Padian, 2008b*; *Padian, 2008c*); as widely-sprawled quadrupeds, limited to rotatory gaits (as defined by *Padian, Li & Pchelnikova, 2010*) and ill-suited to movement on the ground, but possibly adept at climbing (*Unwin, 1987*; *Unwin, 1988*; *Unwin, 1989*; *Unwin, 1999*; *Unwin, 2005*; *Unwin & Bakhurina, 1994*); or as quadrupeds with erect hindlimbs, capable of arboreal locomotion and powerful leaping (*Bennett, 1997b*) or bipedal running (*Padian, 2008a*; *Padian, 2008b*; *Padian, 2008c*).

To some extent, discussions of non-pterodactyloid terrestriality have been intertwined with debates over pterosaur bipedality, as many of the foundations of this hypothesis were set using non-pterodactyloids (*Padian, 1983a*; *Padian, 1983b*). Pterosaur bipedality has always been controversial (*Bennett, 1997a*, p. 107) and has inspired numerous analyses. Most have suggested that habitual bipedalism—either bird-like or otherwise—is unlikely for any pterosaur. Criticisms of this concept include all pterosaurs having a centre of gravity situated towards the shoulders (*Wellnhofer, 1988*; *Bennett, 1997a*; *Sangster, 2003*; *Wilkinson, 2008*); a pedal morphology ill-suited to digitigrady (*Bennett, 1997a*; *Clark et al., 1998*); lever arms of proximal hindlimb musculature which perform poorly at postures imposed by bipedality (*Fastnacht, 2005*; *Costa, Rocha-Barbosa & Kellner, 2014*);

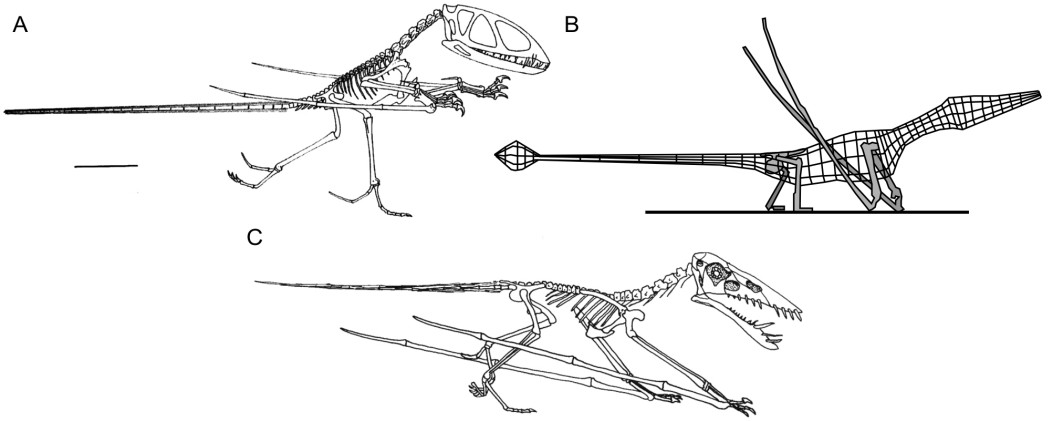

**Figure 1 Select hypotheses for non-pterodactyloid poses made in the last 35 years.** (A) *Padian*'s (*1983a*) bipedal *Dimorphodon macronyx*; (B) redrawn lateral view of the 'Roborhamphus' model discussed by *Unwin (2005)*; (C) quadrupedal *Dorygnathus banthensis* with sprawling forelimbs, reversed from *Padian (2008b)*.

an inability to neatly fold the forelimbs (*Wilkinson, 2008*), and forelimb strength scaling regimes contrasting with those of flying bipeds, but matching those of quadrupeds (*Habib, 2008*). The inability of pterosaur forelimbs to reach the ground has also been disputed (*Unwin, 1996a*; *Bennett, 1997a*), although *Padian* (in *Padian, 2003*; *Padian, 2008a*; *Padian, 2008b*) maintains that the limb proportions of some species, in concert with perceived limited humeral motion at the shoulder, dictates facultative bipedality for some pterosaurs. Functional evidence casting doubt on bipedal postures in pterosaurs is consistent with a wealth of trackway data showing pterosaurs as quadrupedal animals with plantigrade feet (e.g., *Mazin et al., 1995*; *Mazin et al., 2003*; *Lockley & Wright, 2003*; *Hwang et al., 2002*; *Padian, 2003*; *Lockley, Harris & Mitchell, 2008*), and is further bolstered by the unique fit of pterosaur anatomy to these tracks (*Lockley et al., 1995*; *Bennett, 1997a*; *Unwin, 1996a*). Note that recent experimentation with extant crocodilian trackmakers has cast further doubt on perceived similarities between *Pteraichnus* tracks and those of crocodylomorphs (*Kubo, 2008*; contra. *Padian & Olsen, 1984*; *Padian, 2003*).

Of the several interpretations of non-pterodactyloid terrestrial locomotion outlined above, the proposal that they were relatively ineffective terrestrial quadrupeds has gained the largest acceptance (e.g., *Unwin, 1987*; *Unwin, 1988*; *Unwin, 1989*; *Unwin, 1999*; *Unwin, 2005*; *Unwin & Bakhurina, 1994*; *Ősi, 2011*; *Butler, Benson & Barrett, 2013*; *Benson et al., 2014*; *Whyte & Romano, 2014*). The assumed contrast in terrestrial abilities between non-pterodactyloids and pterodactyloids has influenced considerations of not only non-pterodactyloid palaeobiology (e.g., lifestyles and diets—see *Unwin, 2005*; *Ősi, 2011*) but also the evolution of Pterosauria as a whole. Some recent workers have considered the origin of pterodactyloids a 'terrestrialisation' of pterosaurs (*Unwin, 2005*; *Butler, Benson & Barrett, 2013*), and a radical evolutionary departure from the primarily scansorial and volant habits used by earlier members of the group.

The concept of grounded non-pterodactyloids as poor terrestrial locomotors relies on three oft-repeated hypotheses. The first concerns the expansive uropatagium which

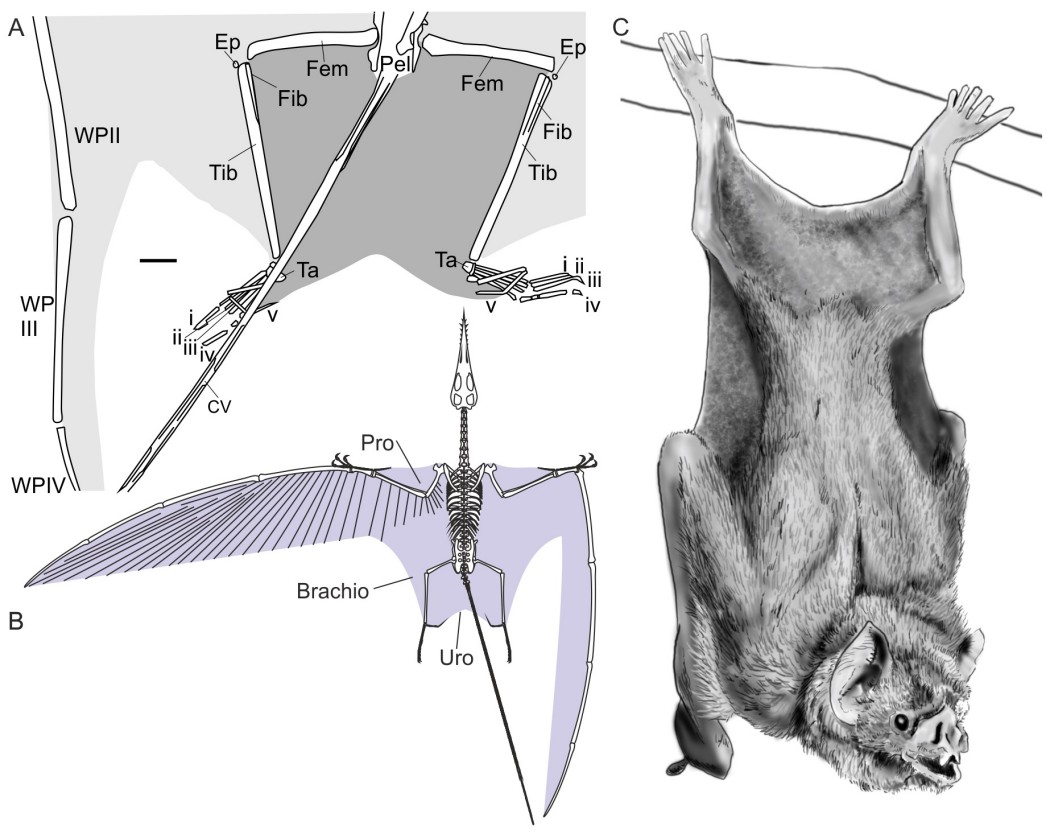

**Figure 2  Pterosaur and bat uropatagia compared.** (A) line drawing of hindlimb region of *Sordes pilosus* specimen PIN 2885/3, showing extensive, toe-supported uropatagium (dark shading) and associated brachiopatagia (light shading); (B) skeletal reconstruction of *Rhamphorhynchus muensteri* showing distribution of membranes in non-pterodactyloid pterosaurs based on fossil remains (see *Elgin, Hone & Frey, 2011*); (C) line drawing of hanging common vampire bat *Desmodus rotundus*, a terrestrially-competent species with an extensive uropatagium analogous to those of non-pterodactyloid pterosaurs. Note *Desmodus* has a small uropatagium compared to other, terrestrially-adept bat species. Scale bar of (A) represents 10 mm, other images not to scale. Abbreviations: Brachio, brachiopatagium; CV, caudal vertebrae; Ep, epiphysis, Fem, femur; Fib, fibula; Pel, pelvis; Pro, propatagium; Ta, tarsals; Tib, tibia; Uro, uropatagium; WP, wing phalanx (numerals denote phalanx number); i–v denote pedal digit numbers. (A) modified from *Unwin & Bakhurina (1994)*; (B) modified from *Witton (2013)*; (C) redrawn from photograph in *Nowak (1994)*.

extended between the hindlimbs of non-pterodactyloids, supported distally by long fifth pedal digits (Fig. 2; *Sharov, 1971*; *Unwin & Bakhurina, 1994*; *Wild, 1994*; *Kellner et al., 2010*). This is reasoned to have restricted independent hindlimb motion and stride length, limited speed and agility, and hindered movement through complex, vegetated environments (e.g., *Unwin, 1988*; *Unwin, 1999*; *Unwin, 2005*; *Ősi, 2011*). The reduction of fifth toe length in pterodactyloids is interpreted as signifying the loss or reduction of this membrane, as evidenced by a pterodactyloid specimen with reduced hindlimb membranes lacking medial contact (*Wellnhofer, 1987*). This 'decoupling' of the hindlimbs from one another is thought to have permitted longer strides and more effective, faster terrestrial locomotion in pterodactyloids, explaining their relative success in inland settings over their ancestors (e.g., *Unwin, 1988*; *Unwin, 1999*; *Unwin, 2005*; *Ősi, 2011*).

The absence of pterosaur trackways from Triassic to Middle Jurassic rocks is a second piece of evidence cited for non-pterodactyloid terrestrial incompetency. The pterosaur body fossil record begins in at least the Norian but, to date, no definitively identified pterosaur trackways occur in rocks pre-dating the Aalenian (*Lockley, Harris & Mitchell, 2008*; *Whyte & Romano, 2014*). The oldest occurrence of pterosaur tracks roughly coincides with the oldest evidence of pterodactyloids (*Andres, Clark & Xu, 2014*; *Whyte & Romano, 2014*) and is seen as evidence for pterosaurs becoming 'terrestrialised', it being assumed that pterodactyloid anatomical nuances allowed exploitation of settings such as tidal flats and lake margins, and creation of a track record (*Unwin, 2005*; *Butler, Benson & Barrett, 2013*). Non-pterodactyloids, by contrast, are assumed so poorly adapted for walking and running that they scarcely used such forms of locomotion, and thus rarely left footprints (*Unwin, 2005*).

Both of these concepts are in keeping with a third hypothesis, that non-pterodactyloids had sprawling forelimbs, and perhaps sprawling hindlimbs as well (*Wellnhofer, 1975*; *Unwin, 1988*; *Unwin, 1999*; *Unwin, 2005*; *Padian, 2008b*). It has been argued that these would limit quadrupedal walking speeds and force reliance on other forms of locomotion—bipedal running or flight—to move rapidly (*Unwin, 1988*; *Unwin, 1999*; *Unwin, 2005*; *Padian, 2008b*). Although some have argued that the hindlimbs of early pterosaurs were erect and powerfully muscled (e.g., *Padian, 1983a*; *Padian, 1983b*; *Padian, 2008b*; *Bennett, 1997b*; *Elgin, Hone & Frey, 2011*), these observations have not influenced some considerations of non-pterodactyloid terrestrial locomotion (*Unwin, 1988*; *Unwin, 2005*). However, even among those proposing erect hindlimbs, the terrestrial prospects of non-pterodactyloids are not considered highly. *Padian (2008a)* and *Padian (2008b)* has suggested that a combination of erect hindlimbs and sprawled forelimbs would incline early pterosaur torsos anteriorly, and, in concert with limited forelimb reach, render them ill-suited to terrestrial locomotion—at least as quadrupeds (this is considered one line of evidence for bipedal habits). In recent years, views that early pterosaurs were inept terrestrial animals have been presented as established and important parts of pterosaur evolutionary history, and said to explain patterns within the pterosaur fossil record (*Unwin, 1999*; *Unwin, 2005*; *Butler, Benson & Barrett, 2013*; *Benson et al., 2014*).

These assumptions have become established despite the low number of dedicated assessments into non-pterodactyloid locomotion. There have been considerably fewer studies into non-pterodactyloid functionality than there are for pterodactyloids, and particularly so in recent years. This probably reflects the larger amount of material available for studies into pterodactyloid mechanics: along with footprints and tracks, many pterodactyloids are known from three-dimensional material which lends itself better to functional studies than the mostly flattened and fragmentary remains forming the non-pterodactyloid record. Nevertheless, some non-pterodactyloid anatomies are well enough known to permit evaluation of arguments suggesting poor terrestriality in these early forms. This is attempted here, with the three principle hypotheses underlying most assessments of non-pterodactyloid terrestriality being considered:

1. Did the large uropatagium of non-pterodactyloids restrict hindlimb function during terrestrial locomotion?

2. Is the absence of non-pterodactyloids trackways related to their terrestrial capabilities?

3. Were the limbs of non-pterodactyloids sprawled during terrestrial locomotion?

## MATERIALS AND METHODS

### Systematic declaration

Pterosaur systematics, and particularly those of early taxa, are currently highly controversial. With so little agreement on multiple aspects of early pterosaur phylogeny including clade content, group definitions, and appropriate nomenclature (e.g., *Unwin, 2003*; *Kellner, 2003*; *Wang et al., 2009*; *Kellner, 2010*; *Dalla Vecchia, 2009*; *Lü et al., 2010*; *Lü et al., 2012*; *Witton, 2013*; *Andres & Myers, 2012*; *Andres, Clark & Xu, 2014*), accurate discussion of pterosaur systematics requires regular citation of the specific taxonomy being followed (e.g., *Andres & Myers, 2012*) or frequent mentions of conflicting phylogenies (e.g., *Witton, 2013*). Neither approach is practical or makes for compelling reading. Thus, unless otherwise stated, this paper uses the nomenclature and taxonomy of the non-pterodactyloid phylogeny of *Lü et al. (2012)*. *Dalla Vecchia (2009)*, *Wang et al. (2010)* and *Andres & Myers (2012)* offer alternative contemporary schemes.

### Material

A number of specimens inform the discussion provided here, but key material includes three-dimensionally preserved remains of *Dimorphodon macronyx*, a well-known Sinemurian, Liassic non-pterodactyloid from Dorset, UK. Observations were chiefly made on the holotype NHMUK R1034, a partial skeleton, and the near complete skeleton NHMUK 41412-13. Both specimens, although partially embedded in matrix, are largely three dimensionally preserved and sufficiently prepared to appreciate most aspects of limb girdle and limb anatomy, especially when viewed in concert with other, less complete *Dimorphodon* material in the Natural History Museum, London. Additional study was made on a near-complete three-dimensional rhamphorhynchine scapulocoracoid from the Callovian-Oxfordian Oxford Clay, UK, NHMUK R5672. *Wellnhofer (1975)* referred this specimen to *Rhamphorhynchus* sp., but diagnostic characters for this genus are presently only known in the skull anatomy and limb proportions of this genus (*Bennett, 1995*). While undoubtedly *Rhamphorhynchus*-like, NHMUK R5672 is conservatively considered an indeterminate rhamphorhynchine here, echoing taxonomic suggestions by *Unwin (1996b)*.

## RESULTS

1. *Did the large uropatagium of non-pterodactyloids restrict hindlimb function during terrestrial locomotion?*

The inference that relatively large uropatagia impeded early pterosaur terrestrial habits has received no detailed evaluation, despite its confident presentation in some literature. ("There can be no doubt that this shackling of the limbs must have hindered pterosaurs as they sought to move around on the ground"—*Unwin, 2005*, p. 204.) It

might be presumed that attributes of fossil pterosaur soft-tissues or observations on modern animals with similar membrane structures support this assertion, but it is only the relatively large size of early pterosaur uropatagia which is cited in favour of this idea (e.g., *Unwin & Bakhurina, 1994*; *Unwin, 2005*). While it is difficult to evaluate the effects of soft-tissues on non-pterodactyloid hindlimb kinematics in the absence of footprints, evidence from pterosaur body fossils, and the anatomy and behaviour of modern animals, conflict with proposals that expansive uropatagia impeded early pterosaur terrestriality.

Many gliding and flying mammals possess large, hindlimb-spanning uropatagia comparable in size to those of non-pterodactyloids (Fig. 2). A number of these species are terrestrially proficient (e.g., *Sollberger, 1940*; *Nowak, 1994*; *Stafford, Thorington Jr & Kawamichi, 2003*; *Riskin et al., 2006*; *Meijaard, Kitchener & Smeenk, 2006*), some spending considerable amounts of time on the ground in pursuit of food or refuge using fast, complex and sometimes strenuous behaviours (*Sollberger, 1940*; *Daniel, 1976*; *Nowak, 1994*; *Pyare & Longland, 2002*; *Riskin et al., 2006*). These animals are not confined to barren habitats, predator-free environments or the result of reduced competition from other terrestrial creatures. Rather, they inhabit complex, predator-filled habitats and have persisted for many millions of years in some regions (*Hand et al., 2009*). Examples include the New Zealand lesser short-tailed bat, *Mystacina tuberculata*, which is reported as having "rodent-like agility on the ground and on trunks, branches, and kiekie vines" by *Daniel* (*1976*; p. 397). Common vampires, *Desmodus rotundus*, rely on their terrestrial skills to stealthily stalk hosts or quickly evade danger using forelimb-propelled galloping (*Nowak, 1994*; *Riskin & Hermanson, 2005*; *Riskin et al., 2006*). Flying squirrels, such as *Glaucomys* species, forage on the ground, are capable of running, and have membranes resilient to frequent digging for fungal food sources (*Sollberger, 1940*). Similarly, membranes of *Mystacina* bats withstand crevice-crawling, as well as digging (*Daniel, 1979*). Clearly, the grounded activities of these animals are not impeded by their patagia, nor do their membranes snag on obstacles or become easily damaged. Presumably, membrane elasticity plays a role in reducing impedance to terrestrial activity, both allowing the limbs to move freely as well as drawing the membranes close to the body to prevent interference with the environment. The extent of such membrane shrinkage can be extreme, rendering them almost indiscernible in some circumstances (*Meijaard, Kitchener & Smeenk, 2006*). Critically, while some membrane-bound extant animals are poor terrestrial locomotors, this has not been linked to membrane size or distribution, but instead to aspects of skeletal morphology, limb strength or myology (*Riskin, Bertram & Hermanson, 2005*).

Certain bats and flying squirrels show that large uropatagia do not rule out terrestrial potential in volant mammals, but are they suitable models for pterosaurs? Fossils of pterosaur wing membranes suggest some similarities to those of modern volant mammals in that they were likely elastic in their proximal regions. Pterosaur brachiopatagia were stiffened by structural fibres distally, but other membrane components—including the uropatagium—lack rigid structural fibres and are widely

considered to have been compliant (e.g., *Padian & Rayner, 1993*; *Unwin & Bakhurina, 1994*; *Bennett, 2000*; *Frey et al., 2003*). *Unwin & Bakhurina (1994)*, describing the uropatagium of *Sordes pilosus*, comment specifically on this, stating "…adjacent to the body the [structural] fibres are shorter, more sinuous and loosely packed, indicating that the propatagium, uropatagium and proximal regions of the cheiropatagium were somewhat softer and more elastic" (p. 64). From this, it can be expected that all pterosaur membranes would contract significantly when the limbs were not extended to flight position, as occurs in many volant mammals, clearing them of obstacles and permitting stretching of the membranes during walking or running. Some evidence for this contraction may be seen in pterosaur fossils with preserved membranes (*Elgin, Hone & Frey, 2011*). Trackways made by running pterodactyloids indirectly demonstrate how elastic their proximal membranes must have been, allowing track makers to take strides of considerable magnitude (*Mazin et al., 2003*) despite membranes stretching from the distal hindlimb to their hands (*Elgin, Hone & Frey, 2011*). The expansion and contraction of brachiopatagia in running pterodactyloids was probably no greater than that experienced by non-pterodactyloid uropatagia during terrestrial activity.

Even if the hindlimb strides of non-pterodactyloids were restricted by membranes, they were likely capable of circumventing this issue by using asymmetrical, bounding gaits (*Witton & Habib, 2010*; *Witton, 2013*; *Hyder, Witton & Martill, 2014*). Indeed, both the fore- and hindlimbs of pterosaurs have been noted for their strength and leaping potential (*Padian, 1983a*; *Bennett, 1997b*; *Habib, 2008*; *Witton & Habib, 2010*), and there are obvious parallels between forelimb-dominated *Desmodus* galloping and recent, compelling hypotheses concerning forelimb use in pterosaur launch (*Habib, 2008*). Pterosaurian bounding locomotion may be countered by exclusive trackway evidence for symmetrical gaits in pterodactyloids (e.g., *Stokes, 1957*; *Mazin et al., 1995*; *Mazin et al., 2003*; *Lockley & Wright, 2003*; *Hwang et al., 2002*), but it remains unclear if these gaits were employed by all pterosaurs, all the time, nor is it clear if interpretations of these tracks are applicable to non-pterodactyloids. Bounding gaits are at least tenable from a functional and biomechanical perspective.

In light of these observations, the proposal that early pterosaurs were terrestrially hindered by their membranes is peculiar. It relies on the uncertain assumption that the uropatagium was especially restrictive compared to other pterosaur wing membranes and behavioural restrictions—membranes snagging on obstacles and limiting stride length—which have no precedent among modern pterosaur analogues. Clear evidence demonstrating broad uropatagia were barriers to early pterosaur terrestriality has yet to be presented, whereas what we know of pterosaur soft-tissues and modern animals with similar anatomy indicates that their membranes likely had little, if any, impact on terrestrial potential.

2. *Is the absence of non-pterodactyloids trackways related to terrestrial capabilities?*

The view that a lack of early pterosaur trackways must equate to their terrestrial ineptitude (e.g., *Unwin, 2005*; *Butler, Benson & Barrett, 2013*) relies on a very literal interpretation of the pterosaur fossil record and an assumption that we can distinguish

genuine absences of fossil phenomena from biases affecting fossil datasets. There are reasons to consider both these assertions uncertain.

The non-pterodactyloid body fossil record is not only poorer than that of pterodactyloids, but also many contemporary terrestrial tetrapod groups (e.g., *Benton & Spencer, 1995*; *Kielan-Jaworowska et al., 2004*). It is particularly impoverished in terrestrial basins (*Butler, Benson & Barrett, 2013*). This is thought to reflect the general lack of inland or near-shore pterosaur-bearing Lagerstätten before the Late Jurassic; the small body sizes and low preservation potential of early pterosaurs; a possibly restricted distribution of the group in its early history; or perhaps existence of the first pterosaurs in habitats unconducive to fossilisation and sediment accumulation—inland forests or upland environments (*Bennett, 1997b*; *Unwin, 2005*; *Witton, 2013*; *Butler, Benson & Barrett, 2013*). Regardless of the cause, recent studies have concluded that recorded patterns of Triassic and Jurassic pterosaur diversity—the interval dominated by non-pterodactyloids—have little statistical significance (e.g., *Butler, Benson & Barrett, 2013*; *Upchurch et al., 2014*), and that our understanding of early pterosaur history remains generally poor. This is difficult to reconcile with suggestions that the lack of early pterosaur fossils—specifically their track record—is somehow significant. If understanding of the early pterosaur record is demonstrably limited, how can any apparent trends or patterns in that data be confidently interpreted, and especially those reliant on an absence of data?

It seems unwise to link the absence of a track record to a very specific cause, such as functional anatomy when there are a number of reasons why non-pterodactyloids may not have an ichnological record. If non-pterodactyloids were genuinely rare in terrestrial basins—as their record currently indicates—their likelihood of creating traces must also be low. Likewise, it seems most early pterosaurs were small, with wingspans of 1–2 m (*O'Sullivan, Martill & Groocock, 2013*) and corresponding masses of 0.55–3.26 kg (using data from *Witton, 2008*). Their footprints would thus be small and shallow, without substantial underprinting, and require exceptional conditions for impression, fossilisation and discovery. In contrast, pterodactyloids are generally larger bodied than early pterosaurs (*Hone & Benton, 2007*; *Benson et al., 2014*), which may constitute creation of deeper, longer-lasting tracks which are better suited to fossilisation and detection. A related problem concerns our ability to distinguish the footprints of pterodactyloids from those expected of non-pterodactyloids (*Lockley, Harris & Mitchell, 2008*): all pterosaurs have the same basic manus and pes structure, the only exception being the longer fifth toe in non-pterodactyloids. Given the role of this structure in supporting the uropatagium, it may have been held aloft when walking (*Lockley, Harris & Mitchell, 2008*). If so, the tracks of all pterosaurs might look similar, and some alleged Jurassic pterodactyloid ichnites may be misidentified.

It should also not be assumed that early pterosaurs and pterodactyloids occupied ecologies with similar track-making potential. The start of the pterosaur footprint record in the Middle Jurassic roughly corresponds with the emergence of pterodactyloid clades predicted to be waders, suspension-feeders and molluscivores (ctenochasmatoids and dsungaripterids—*Unwin, 2005*; *Witton, 2013*). Such animals are expected to

routinely patrol lake margins and other habitats suitable to footprint preservation in search of food. *Lockley & Wright (2003)* and *Lockley, Harris & Mitchell (2008)* note that pterodactyloid tracks are frequently associated with invertebrate traces and occasional feeding marks, which may indicate foraging was a common factor in pterosaur ichnite creation, inferring ecological influences on the delayed start of the pterosaur ichnological record. By contrast, non-pterodactyloids are largely perceived as pelagic piscivores or insectivores (*Wellnhofer, 1975*; *Wild, 1978*; *Chatterjee & Templin, 2004*; *Ősi, 2011*; *Witton, 2008*; *Witton, 2013*), neither of which are habits lending themselves to sustained terrestrial activity on mudflats, water margins or other settings liable to preserving footprints.

Perhaps most importantly, early pterosaurs are not alone in having a very sparse track record. The tracks and traces of many fully terrestrial Mesozoic clades are surprisingly poorly known—examples include geographically widespread, long-lived lineages with good body fossil records, such as Mesozoic Mammaliaformes, tyrannosaurids and ceratopsids (*Lockley & Hunt, 1995*; *Kielan-Jaworowska et al., 2004*; *McCrea et al., 2014*). Not only are the ichnological records of these groups poor—restricted to single localities in some cases—but many ichnites referred to them are controversially identified (*Kielan-Jaworowska et al., 2004*; *McCrea et al., 2014*). This occurs despite these animals seemingly being abundant (as evidenced by their good body fossil records) and fully terrestrial in their habits, thus potentially creating tracks in virtually all of their activities (unlike pterosaurs, which, in being volant, avoided track creation much of the time). In contrast to perceptions of the pterosaur track record however, the sparse trackways of Mesozoic Mammaliaformes or certain dinosaur clades are not interpreted as signs terrestrial ineptitude, but as biases of behaviour, ecology, preservation, sampling or interpretation.

Ultimately, while the absence of early pterosaur footprints is an intriguing phenomenon of the pterosaur record, and one with possible implications for the development of terrestriality in Pterosauria, its significance cannot be divorced from a number of factors unrelated to functional morphology. As with any case supported by negative evidence, data deficits can only be interpreted so far, especially when related datasets are demonstrably poor. Considering the absence of early pterosaur tracks as significant requires ignorance of not only statistics on the quality of the pterosaur fossil record, but also data concerning early pterosaur palaeobiology and the broader ichnological record. Other sources of evidence should be pursued for more reliable insights into the development of pterosaur terrestriality.

3. *Were the limbs of non-pterodactyloids sprawled during terrestrial locomotion?*

Postural sprawl and the use of rotatory limb mechanics has been proposed for grounded non-pterodactyloids from assessments of their limb joint arthrology (e.g., *Wellnhofer, 1975*; *Unwin, 1988*; *Unwin, 1999*; *Unwin, 2005*; *Padian, 2008b*). These suggestions have mostly applied to their forelimbs, but some have suggested that both limbsets were constrained to sprawling stances (*Unwin, 1988*; *Unwin, 1999*; *Unwin, 2005*). *Unwin (1988)* argued that the *Dimorphodon* femoral-pelvic joint projected the femur anterolaterally and somewhat dorsally when 'naturally articulated', while the tibiotarsus was capable of twisting medially at the knee, permitting the foot

to face forwards. This is said to allow for semi-erect or sprawling stances, which are in accordance with suggested similarities between the pelves of *Dimorphodon* and the sprawling or semi-erect archosauriform *Euparkeria capensis* (*Unwin, 1988*). Computer modelling has also predicted entirely sprawling stances and rotatory gaits for non-pterodactyloids through a digital model of *Rhamphorhynchus* (Fig. 1B; *Unwin, 2005*). The methodology behind this has not been presented, but the resultant digital non-pterodactyloid model 'Roborhamphus' shows hindlimbs projecting entirely laterally from the body, similarly-sprawling forelimbs, low clearance from the ground and slow walking speeds (*Unwin, 2005*). The latter is seemingly a consequence of the limited reach afforded by the sprawling limbs.

There are several reasons to think that the non-pterodactyloid hindlimb did not sprawl. Firstly, the assumption that a 'natural articulation' of the hindlimb can be determined from acetabulum and femoral head morphology (*Unwin, 1988*) is problematic. As evidenced by debates over 'osteological neutral pose' in fossil animal necks (e.g., *Stevens & Parrish, 1999*; *Taylor, Wedel & Naish, 2009*; *Taylor & Wedel, 2013*; *Stevens, 2013*), attempts to determine 'neutral' or 'natural' poses of animal joints rely on arbitrary assignments of optimal joint configurations which often have little or no significance to typical animal postures (*Taylor, Wedel & Naish, 2009*). It is probably unwise to suggest the hindlimb of *Dimorphodon* sprawled based on acetabulum and femoral head morphology alone.

Secondly, the pelves of *Dimorphodon* and other early pterosaurs are clearly distinguished from those of *Euparkeria* and other sprawling animals in having a well-developed preacetabular process (*Unwin, 1988*; *Hyder, Witton & Martill, 2014*). In this respect, non-pterodactyloid pelves resemble those of other ornithodirans— including pterodactyloids—and mammals. These taxa are characterised by erect limbs, the preacetabular process anchoring large hip flexors for moving the hindlimb forward in the parasagittal plane (*Hyder, Witton & Martill, 2014*). Assessments of pterosaur hindlimb muscle mechanics seem to confirm that the pterosaur pelvic and femoral musculoskeletal system is optimally configured for an erect stance (*Fastnacht, 2005*; *Costa, Rocha-Barbosa & Kellner, 2014*). Furthermore, while arguments for bipedal, pronograde pterosaurs with parasagittal hindlimbs and digitigrade pedes (*Padian, 1983a*; *Padian, 1985*) have been largely criticised in recent years (e.g., *Wellnhofer, 1988*; *Bennett, 1997a*; *Clark et al., 1998*; *Fastnacht, 2005*—also see above), observations that their hip, knee and ankle articulations have hallmarks of upright limb functionality have been borne out by further study (*Bennett, 1997b*; *Padian, 2008a*; *Padian, 2008b*; *Fastnacht, 2005*; *Costa, Rocha-Barbosa & Kellner, 2014*).

Thirdly, virtually all recent models of pterosaur evolution suggest taxa with erect hindlimbs bracket non-pterodactyloids, with *Scleromochlus taylori* and non-pterosaurian ornithodirans on one side, and pterodactyloids the other (*Sereno, 1991*; *Benton, 1999*; *Hone & Benton, 2008*; *Nesbitt, 2011*; but also see *Bennett, 2013*). This implicates erect hindlimb postures as probably ancestral for Pterosauria and, given the similarity of their pelvic and hindlimb osteology to their nearest probable relatives,

there is little reason to assume non-pterodactyloids deviated from this ancestral state (*Bennett, 1997b*; *Padian, 2008a*; *Hyder, Witton & Martill, 2014*). It seems that multiple lines of evidence indicate erect hindlimbs across Pterosauria, including all known non-pterodactyloids.

Relatively little has been said on the stature of non-pterodactyloid forelimbs. Traditionally, they have been reconstructed as sprawling. *Wellnhofer (1975)* observed that the glenoid of *Rhamphorhynchus muensteri* projected laterally and permitted anterodorsal and posteroventral motion of the humerus, but that anterior and posterior motion was limited, and that the humerus could not be adducted below the level of the scapula. *Padian (1983a)* suggested that the glenoid of *Dimorphodon* permitted a 90°arc of rotation, most of it dorsal to the glenoid, and later suggested the shoulder joint of *Dorygnathus* permitted little movement below the frontal plane (Fig. 1C; *Padian, 2008b*). This is said to limit *Dorygnathus* to a sprawling forelimb stance during quadrupedal locomotion which could not match pace with the erect hindlimbs during running, for which bipedality was employed (*Padian, 2008b*). This configuration, which *Padian (2003)* and *Padian (2008a)* considers typical of all 'basal pterosaurs' (presumably non-pterodactyloids), is also thought to limit ventral reach of the forelimb to the extent that bipedal locomotion must be used, as the pectoral region is depressed significantly below that of the hindlimb in a state considered awkward for effective locomotion (*Padian, 1983b*; *Padian, 2003*; *Padian, 2008a*; *Padian, 2008b*). *Unwin (2005)* showed the digital 'Roborhamphus' forelimbs in a sprawling fashion somewhat consistent with these models (Fig. 1B).

Notions that non-pterodactyloid forelimbs were confined to sprawling stances by their glenoids are based in part on the pectoral girdles of rhamphorhynchine pterosaurs (e.g., *Wellnhofer, 1975*; *Padian, 2008b*). The glenoids of these animals are typified by the isolated, but well-preserved scapulocoracoid of an Oxford Clay rhamphorhynchine, NHMUK R5672 (Figs. 3D–3G and 3I). Here, the glenoid is a laterally prominent structure with a long axis aligned with the base of the scapula. The anterior and posterior ends are bordered by a prominent lower tubercle and supraglenoidal buttress, respectively, between which occurs a deeply-curved, saddle-shaped articular surface. This wraps almost 90° from the lateral face to the dorsal, suggesting ample humeral motion lateral and dorsal to the glenoid. The articular face is anteroposteriorly broadest in its dorsal region and most constrained laterally. The ventral extent of the glenoid is marked by a laterally-projecting ridge between the posterior buttress and anterior tubercle. This ridge is continuous with the lower extent of the scapula, supraglenoidal buttress and lower tubercle so that the ventral face of the glenoid is a wide, flat surface instead of a saddle shaped-joint like that of the dorsal region. As noted by previous authors (e.g., *Wellnhofer, 1975*; *Padian, 2008a*; *Padian, 2008b*), such glenoids clearly did not permit humeral adduction below the level of the scapula, and likely limited fore- and aft-motion of the humerus at maximal adduction. Humeral motion was likely less constrained dorsally, however. Given their marked dorsoventral asymmetry, these glenoids are hereafter referred to as 'asymmetric'.

A survey of non-pterodactyloid remains suggests asymmetric glenoids occur in a number of taxa, including the Jurassic rhamphorhynchines *Rhamphorhynchus muensteri*

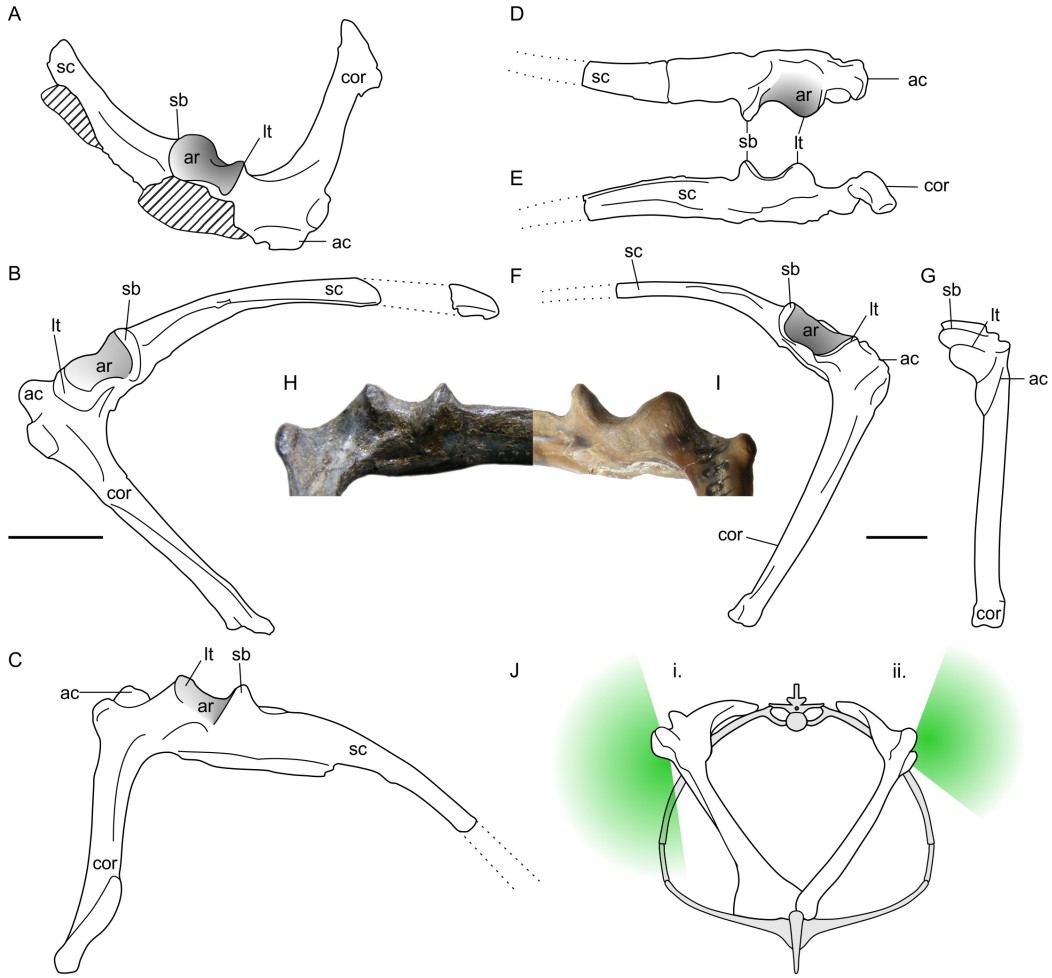

**Figure 3 Non-pterodactyloid glenoid morphology.** (A–C) line drawings of NHMUK R1034 *Dimorphodon macronyx* left scapulocoracoid in anterodorsal (A), lateral (B) and ventrolateral (C) aspect; (D–G), NHMUK R5672, indeterminate rhamphorhynchine right scapulocoracoid in dorsal (D), ventral (E), lateral (F) and anterior (G) aspects; (H) photograph of the NHMUK R1034 glenoid ('symmetric' morph), in posteroventral aspect; (I) photograph of NHMUK R5672 ('asymmetric' morph) in posteroventral aspect; (J) schematic reconstruction of a non-pterodactyloid torso with a symmetric (i) and asymmetric (ii) glenoid conditions, where green shading approximates articulatory range of the humerus in the vertical plane based on extent of articular surface. Scapulocoracoids in (J) reconstructed based on specimens illustrated herein and models of pectoral anatomy presented for other early pterosaurs (*Wellnhofer, 1975*; *Wellnhofer, 1991*; *Bennett, 2003*). Note this is only approximate for *Dimorphodon* because its sternum remains unknown. ac, acromion process; ar, articular face of glenoid; cor, coracoid; lt, lower tubercle; sc, scapula; sb, supraglenoidal buttress. Scale bars represent 10 mm. (J) modified from *Wellnhofer (1991)*.

(see numerous examples in *Wellnhofer, 1975*); the recently-named Kimmeridge Clay *Rhamphorhynchus etchesi* (MJML-K1597, *O'Sullivan & Martill, in press*); *Dorygnathus banthensis* (GPIT 1645/1, *Padian, 2008b*); and *Sericipterus wucaiwanensis* (IVPP V14725, *Andres, Clark & Xing, 2010*). They thus appear to be typical for rhamphorhynchines, and further occurrences may occur within Campylognathoididae, including the Triassic *Eudimorphodon ranzii* (MCSNB 2888, *Wild, 1978*), and Jurassic *Campylognathoides liasicus*

(SMNS 11879, *Padian, 2008c*). Asymmetric glenoids may be more widely spread across non-pterodactyloids than this, but establishing their frequency is complicated by a deficit of good preservation and frequently unmet requirements for particular scapulocoracoid orientations in flattened specimens. Accordingly, the shape of the glenoid cannot be established for many taxa which may otherwise be considered well-known, such as the Anurognathidae.

Asymmetric glenoids are not the only shoulder morph of non-pterodactyloids, how-ever: the well-preserved glenoids on *Dimorphodon macronyx* specimens NHMUK R1034 and 41412-13 are rather differently constructed (Figs. 3A–3C and 3H). Although following the same basic configuration as other non-pterodactyloids, these specimens have a larger supraglenoidal buttress which projects further ventrally than the lower tubercle. The dorsal portion of the glenoid articular face is similar to that described above, but the ventral por-tion wraps onto the underside of the glenoid until it meets the scapula shaft, instead of ter-minating at an elevated scapular margin. This gives the glenoid a spool- or hourglass-like appearance in lateral view, and presents no obvious restriction to adducting the humerus to a subvertical position. As with the rest of the glenoid, the ventral articular region is widely open anteroposteriorly (although not as much as the dorsal region) and is estimated to permit 90° of anteroposterior humeral rotation beneath the body. This observation contrasts with previous assessments of *Dimorphodon* glenoid morphology, which sug-gested an anatomy and arthrological range akin to the 'asymmetric' morph outlined above (*Padian, 1983b*; *Padian, 2003*; *Padian, 2008b*). As demonstrated in Figs. 3H and 3I, the ventral morphologies and likely arthrological ranges of these glenoids are quite distinct.

Glenoids like those seen in *Dimorphodon* are hereafter referred to as 'symmetric', after their relatively similar dorsal and ventral articular surfaces. It seems such glenoids were rare in non-pterodactyloids: other than *Dimorphodon*, only the wukongopterids *Darwinopterus linglongtaensis* (IVPP V16049, *Wang et al., 2010*) and *Darwinopterus robustodens* (HGM 41HIIII-0309A; *Lü et al., 2011*) seem to possess them, although the caveats mentioned above mean this assessment should not be considered definitive. It is notable that some aspects of pterodactyloid glenoids are similar to this 'symmetric' condition, including the relatively large supraglenoidal buttress, expansive articular face, and absence of an arthrologically prohibitive ventral margin.

The two non-pterodactyloid glenoid morphologies identified here have different implications for adoption of sprawling or erect postures. Asymmetric glenoids seem to obligate forelimb sprawling, whereas symmetric glenoids could permit either sprawling or upright limb usage. As noted above, relying on a single joint for insight into animal postures can be misleading, and using only glenoid shape to infer forelimb postures in non-pterodactyloids may be unwise. Additional insights on the stances of these animals are afforded by aspects of their distal humeri, however. The morphology of distal humeri seems characteristic of stance in extant quadrupeds, and has been used to predict sprawling or erect limb carriage in extinct animals (*Fujiwara & Hutchinson, 2012*). This is possible because the size of osteological correlates of wrist flexor and extensor muscles, as well as those of elbow extensors, provide insights into primary mechanical loads placed on

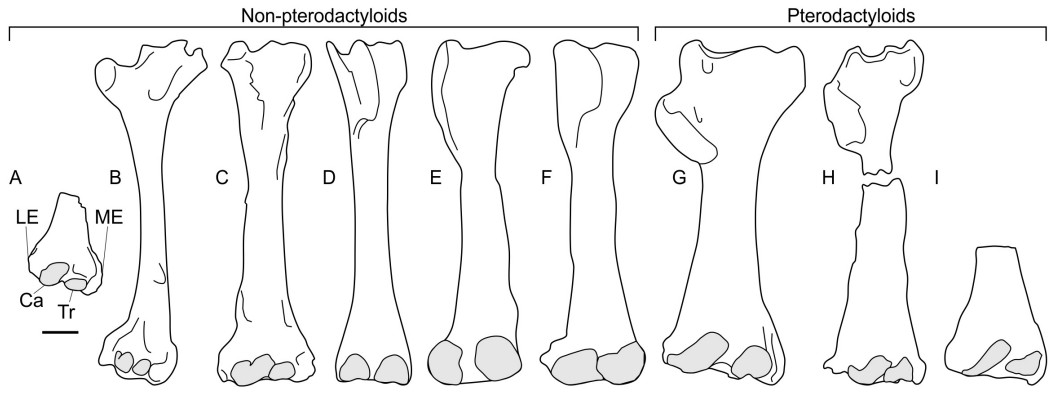

**Figure 4 Pterosaur humeri in anterior view, showing development of lateral and medial epicondyles adjacent to the capitula and trochleae (shaded grey) in non-pterodactyloids (A–F) and pterodactyloids (G–I).** (A) NHMUK 42016, *Dimorphodon macronyx*; (B) YPM 350 (F) *Di. macronyx*; (C) JPM04-0008, *Archaeoistiodactylus linglongtaensis*; (D) *Wellnhofer*'s (*1975*) *Rhamphorhynchus muensteri* humerus; (E) SMNS 51827, *Dorygnathus banthensis*; (F) SMNS 50164, *Do. banthensis*; (G) YPM 1164, *Pteranodon* sp; (H) MOR 691, *Montanazhdarcho minor*; (I) IVPP V.2777, *Dsungaripterus weii*. Note the relatively poorly developed epicondyles in (D–F) and how the distal humeri of (A–C) resemble those of pterodactyloids more than other non-pterodactyloids. Ca, capitulum; LE, lateral epicondyle; ME, medial epicondyle; Tr, trochlea. Scale bars represent 10 mm, except for (G) and (H), which equal 50 mm. (B) after *Padian (1983a)*; (D) after *Wellnhofer (1975)*; (E) after *Padian (2008b)*; (G) modified from *Bennett (2001)*; (H) after *McGowen et al. (2002)*; (I) after *Young (1964)*. (D), (E) and (H) are reversed from their sources to enhance comparability.

the distal humerus and, therefore, an insight into habitual forelimb postures (*Fujiwara & Hutchinson, 2012*). This method, grounded and tested in a biometric dataset of 318 living taxa, has obvious utility for fossil species where interpreting limb posture based on arthrology alone can be controversial. *Fujiwara & Hutchinson (2012)* have already applied their technique to a pterodactyloid (*Anhanguera piscator*) humerus and found it met expectations of animals using an upright posture, agreeing with other predictions made from limb bone arthrology and trackway data for erect forelimb use in pterodactyloids (e.g., *Unwin, 1996a*; *Bennett, 1997a*; *Mazin et al., 2003*). *Fujiwara & Hutchinson (2012)* note that their conclusions likely apply to other pterodactyloids, which have broadly similar humeri to *Anhanguera* (Figs. 4G–4I), implying that occurrences of equivalent humeral morphology in other pterosaurs might suggest similar forelimb use and stance.

A survey of non-pterodactyloid humeri shows variation in their distal ends that correlate with distinctions in their glenoid shapes (Fig. 4). The distal humeri of pterosaurs with asymmetric glenoids possess markedly reduced medial and lateral epicondyles so that, in anterior view, their distal humeri show relatively little expansion from the diaphyseal shafts. For this reason, the palmar aspect of their distal humeri are dominated by the capitula and trochlea (Figs. 4D–4F). This is seen in at least *Rhamphorhynchus* (*Wellnhofer, 1975*) and *Dorygnathus* (SMNS 51827, SMNS 50164, see *Padian, 2008b*), and perhaps also *Eudimorphodon* (MCSNB 2888) and *Campylognathoides* (see examples in *Padian, 2008c*), although the flattened, often oblique preservation of humeri in specimens of the latter taxon prohibits full confidence in this observation (*Wild, 1978*; *Padian, 2008c*). The

lack of well-developed lateral and medial epicondyles in these humeri contrasts with the pterodactyloid condition, in which these structures are prominent and the distal humeri are expanded (Figs. 4G–4F). However, the distal humeri of *Dimorphodon* (NHMUK 42016, YPM 350) and the probable wukongopterid *Archaeoistiodactylus linglongtaensis* (JPM04-0008, see *Sullivan et al., 2014* for comments on the phylogenetic position of this species) are much more pterodactyloid-like. In these humeri, well-developed lateral and medial epicondyles create a splayed distal termination much broader than either the humeral diaphysis or the combined width of the capitulum and trochlea (Figs. 4A–4C, *Padian, 1983a*; *Lü & Fucha, 2010*). In *Dimorphodon* at least, the medial condyle is also distally displaced compared to the lateral. This results in the distal ends of *Dimorphodon* and wukongopterid humeri being clearly different to those of rhamphorhynchines and campylognathoidids, but morphologically very similar to those of pterodactyloids (Figs. 4G–4I).

The correlation between these humeral conditions and glenoid morphology is potentially significant. The similarity of *Dimorphodon* and wukongopterid humeri to those of pterodactyloids implies a similar mechanical regime being experienced at the elbow region which, following *Fujiwara & Hutchinson (2012)* and other studies on pterodactyloid humeral orientation when walking (e.g., *Unwin, 1996a*; *Bennett, 1997a*; *Mazin et al., 2003*), might indicate the employment of upright stances. As noted above, portions of the symmetric glenoid articular surface project ventrally in a manner expected for animals with erect forelimbs. The large, open junction between the ventral articular surface, lower extent of the supraglenoidal buttress and the lateral face of the scapula seems capable—perhaps even well-suited—to bolstering a fully adducted forelimb for standing and walking. These anatomies raise the possibility of wukongopterids and *Dimorphodon* being capable of erect forelimb postures. If, as noted by *Fujiwara & Hutchinson (2012)*, pterodactyloid-like distal humeri correlate with an upright forelimb stance, such postures may even be likely: the assumption that sprawling forelimbs were common to all non-pterodactyloids clearly warrants further investigation. Moreover, the possibility that some early pterosaurs could fully adduct their humeri suggests that reaching the ground in a quadrupedal stance may not have been difficult, as has been proposed (*Padian, 1983b*; *Padian, 2008a*; *Padian, 2008b*).

The humeral morphology in rhamphorhynchine and campylognathoidid non-pterodactyloids differs from those seen in erect-limbed pterodactyloids (*Fujiwara & Hutchinson, 2012*) and also suits their glenoid morphology. Asymmetric glenoids seem to prohibit humeral adduction into an erect stance, and it would be predicted that correlates for a different set of forelimb muscles—likely those suited to sprawling—would be emphasised at the distal humerus compared to those seen in pterodactyloids. Lack of indications of erect poses suggests these pterosaurs fit 'traditional' models of sprawling forelimbs in non-pterodactyloids (e.g., *Wellnhofer, 1975*; *Unwin, 2005*; *Padian, 2008b*). It might be predicted that their stance and walking gaits required relatively little wrist motion, as evidenced by their weakly developed epicondyles for muscle attachment related to carpal operation.

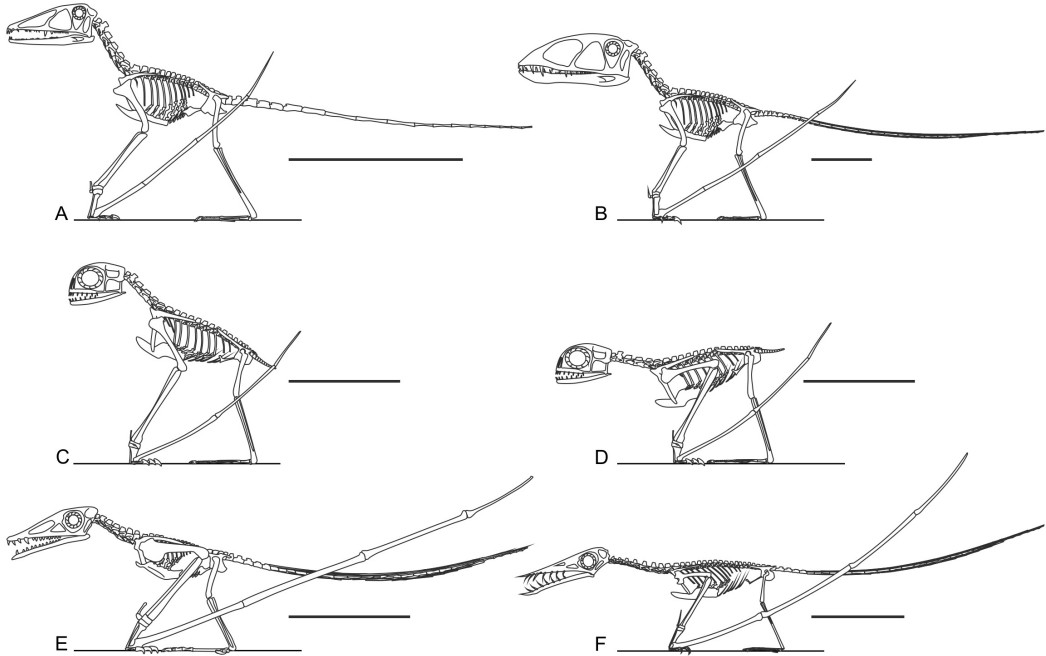

**Figure 5 Skeletal reconstructions of non-pterodactyloid pterosaurs.** Are non-pterodactyloids ubiquitously equipped with short, slender limbs? Skeletal reconstructions of taxa such as *Preondactylus bufarini* (A), *Dimorphodon macronyx* (B) and *Anurognathus ammoni* (C and D, in erect and crouched poses respectively, acknowledging the poorly known glenoid condition of anurognathids) show they have proportionally long, robust limbs. Only some non-pterodactyloids, including the Early Jurassic campylognathoidid *Campylognathoides liasicus* (E) and Late Jurassic rhamphorhynchine *Rhamphorhynchus muensteri* (F) have proportionally short and slender hindlimbs. *Preondactylus* has been reconstructed with erect forelimbs based on its grossly similar humeral morphology to *Dimorphodon*, although it remains to be established that this similarity extends to more detailed forelimb anatomy. Scale bars represent 100 mm, except for (C) and (D), which represent 50 mm. Skeletal reconstructions modified from *Witton (2013)*.

## DISCUSSION

### Other indications of terrestrial competency in non-pterodactyloids

The possibility that some non-pterodactyloids were capable of fully upright stances, and unconstrained during terrestrial locomotion by their membranes, might have broad implications for our perception of their palaeobiology and role in pterosaur evolutionary history. Note, however, that these are not the only aspects of early pterosaur anatomy indicating greater terrestrial potential than generally anticipated.

It has been suggested that non-pterodactyloid limbs are too short and slender for effective terrestrial locomotion (Fig. 5, *Ősi, 2011*). This is probably an over-generalisation: early pterosaur anatomy is quite disparate in many respects (*Witton, 2013*). Several well-known taxa do possess short and/or slender limbs (e.g., Figs. 5E and 5F), but Triassic and Jurassic taxa such as *Dimorphodon*, anurognathids and *Preondactylus bufarinii* possess long, robust, and near-equally sized limbs with well-developed extremities (Figs. 5A–5D; *Owen, 1870*; *Dalla Vecchia, 1998*; *Bennett, 2007*; *Padian, 2008a*). Indeed, the limbs of some non-pterodactyloids are more substantially developed and proportionate than those of

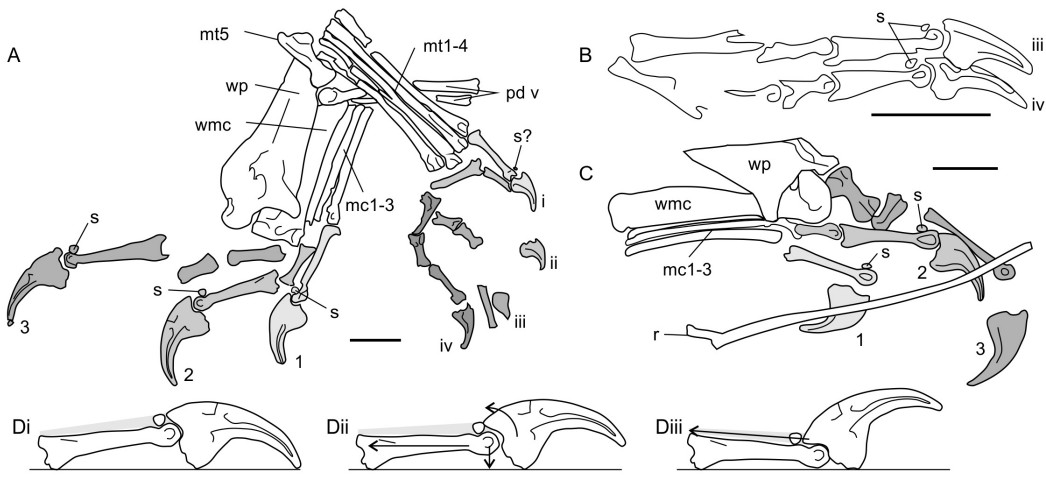

**Figure 6 Antungual sesamoids in pterosaurs.** (A) manus and pes of NHMUK 41212 *Dimorphodon macronyx*; (B) partial pes skeleton of GSM 1546 *Di. macronyx*; (C) manus of BSP 1938 I 49 *Dorygnathus banthensis*; (D) proposed interactions of pterosaur unguals with hard substrates, and utilisation of antungual sesamoids (extensor tendon shown in grey shading). (Di) terminal phalanges of *Dimorphodon* manual digit 2 show as resting on a hard substrate without loading; (Dii) passive hyperextension of the ungual, where pulling or depressing the phalanges (force vectors shown with arrows) retract the ungual to contact the sesamoid; (Diii) active hyperextension of the ungual, where the extensor tendon is pulled to clear the ungual tip of the ground using the additional lever arm length afforded by the sesamoid. (A–C) shading and numbers denote identification of clawed digits (Arabic numerals for manual digits, unary for pedal). Some skeletal elements present on the illustrated specimens are omitted for clarity. mc1-3, metacarpals 1-3; mt1-4, metatarsals 1-4; mt5, metatarsal 5; pd v, pedal digit 5; r, dorsal rib; s, sesamoids; wmc, wing metacarpal; wp, wing finger proximal phalanx. Scale bars represent 10 mm.

seemingly terrestrially-competent pterodactyloids, such as azhdarchids (*Witton & Naish, 2008*). It has been noted that several early pterosaur hindlimb skeletons possess features of subcursoriality (*Padian, 1983b*; *Padian, 2003*; *Padian, 2008a*; *Padian, 2008b*), and this is also true of their forelimbs. Subcursorial features include long limbs relative to their bodies, joints with hinge-like mobility, short and massive propodia, slender and distally reduced/fused fibulae, digitigrade manus and elongate metapodia (see *Coombs Jr, 1978*, p. 399 and 402). It bears repeating that the limbs of pterosaurs—including those of non-pterodactyloids—have been frequently identified as powerfully muscled and strongly built for leaping and flying (*Padian, 1983a*; *Padian, 1983b*; *Padian, 2003*; *Padian, 2008a*; *Padian, 2008b*; *Bennett, 1997b*; *Fastnacht, 2005*; *Habib, 2008*; *Witton & Habib, 2010*), and were therefore likely capable of supporting some grounded activity, perhaps even sustained and energetic terrestrial behaviours. It is likely that the slender, disproportionate limbs of some better known non-pterodactyloids such as *Rhamphorhynchus* have biased opinions on the terrestrial ability non-pterodactyloids as a whole: considered independently, the long, proportionate and robust limbs of genera such as *Dimorphodon*, anurognathids and *Preondactylus* might be viewed as well-suited to terrestrial locomotion.

The digits of several non-pterodactyloid species are also adorned with features which may betray routine terrestrial habits: antungual sesamoids (Fig. 6). These small, round bones are situated on the dorsal surfaces of the penultimate manual phalanges of many Tri-

assic and Lower Jurassic pterosaur specimens, including the Triassic taxa *Eudimorphodon ranzii* (MCSNB 2888), *Carniadactylus rosenfeldi* (MFSN 1797), *Peteinosaurus zambellii* (MCSNB 2887), the "*Eudimorphodon*" specimen MCSNB 8950, the "*Peteinosaurus*" specimen MCSNB 3359 (*Wild, 1978*; *Wild, 1994*; *Dalla Vecchia, 2009*) as well as the Jurassic pterosaurs *Dorygnathus* (e.g., Fig. 6C, BSP 1938 I 49; see also *Padian, 2008b*) and *Dimorphodon* (Fig. 6A, NHMUK 41212; NHMUK R1034; see *Padian, 1983a*; *Unwin, 1988*). *Dimorphodon* is unusual in also bearing pedal antungual sesamoids, spreading their distribution across all clawed digits (Fig. 6B, GSM 1546; *Unwin, 1988*). Antungual sesamoids are present in an osteologically immature specimen of *Eudimorphodon* (MPUM 6009; *Wild, 1978*), suggesting they are not just confined to gerontic, well-ossified adults.

Pterosaur antungual sesamoids are consistently preserved dorsally adjacent to the articular condyles of penultimate phalanges and were presumably situated within the tendons of the digit extensors (*Bennett, 2008*). Their function has not been explored in detail, but two studies (*Unwin, 1988*; *Bennett, 1997b*) cite them as part of a suite of characters important to pterosaur grasping and climbing capabilities. Anatomies related to grasping and climbing are relatively well explored (see *Sustaita et al., 2013* for a recent review) but, to this author's knowledge, extension of the ungual is not generally associated with this behaviour. An exception might be climbing geckos, which retract adhesive pads situated on the distal ends of their digits before each step (*Autumn et al., 2006*; *Russell & Higham, 2009*). However, these geckos famously adhere themselves to substrates via manipulation of molecular forces, not with claws, and their climbing methods are unlikely to mirror those used by pterosaurs.

Antungual sesamoids are currently only known pterosaurs, and terrestrial reptiles: several squamate lineages (*Haines, 1969*; *Jerez, Mangione & Abdala, 2010*; *Otero & Hoyos, 2013*) and the semiaquatic 'bottom walking' Triassic turtle *Proganochelys* (*Gaffney, 1990*). *Gaffney (1990)* proposed that antungual sesamoids confer functions typical of other reptile sesamoids: increasing tendon moment arm lengths around joints, ensuring nutrient delivery to tendons by limiting strain or pressure on joints articulated to their extremes (*Haines, 1969*), or strengthening tendons (*Nussbaum, 1982*). High mechanical stresses on the dorsal side of phalangeal-ungual joints seem to be the most likely catalyst for antungual sesamoid development. Perhaps the only shared functional attributes between pterosaurs, squamates and *Proganochelys* are large unguals and the potential to walk on firm substrates. It may be that these two factors alone can account for antungual sesamoid development. One possibility is that deflection of large, curving unguals by hard surfaces induces pressure on the extensor tendon, promoting the development of a sesamoid to maintain tendon nutrient flow during sustained bouts of standing and walking (Fig. 6Dii). Alternatively, deliberate hyperextension of claws may promote antungual sesamoid development as means to increase the extensor tendon moment arm, and thus improve efficiency of claw retraction (Fig. 6Diii). It is notable that pterosaurs with antungual sesamoids possess expanded, deeply grooved penultimate phalangeal terminations and large ungual extensor tubercles (Figs. 6A–6C), similar to the phalanges of animals with hyperextensible digits, such as cats, dromaeosaurids and schizotherine

chalicotheres (*Coombs, 1983*; *Gonyea & Ashworth, 1975*; *Parsons & Parsons, 2009*). By contrast, pterosaurs lacking antungual sesamoids have relatively small, weakly developed phalangeal-ungual joints (e.g., *Wellnhofer, 1975*; *Clark et al., 1998*), suggesting limited potential for hyperextension. Ungual hyperextension has evolved repeatedly within terrestrial tetrapods to avoid claw blunting (e.g., *Gonyea & Ashworth, 1975*; *Coombs, 1983*) or to release strong grips, *Autumn et al. (2006)* and *Russell & Higham (2009)* has been proposed as an explanation for the lack of ungual traces in some pterodactyloid ichnites (*Frey et al., 2003*).

These hypotheses share frequent ungual interaction with hard substrates as the chief adaptive pressure for antungual sesamoid development. Sustained activity in terrestrial settings is perhaps the most likely cause of this interaction, and congruent with the seemingly-exclusive development of antungual sesamoids in terrestrialised taxa such as squamates and *Proganochelys*. If antungual sesamoids do represent such adaptations, their development in non-pterodactyloids may represent further evidence of terrestrial habits in early pterosaurs.

## The terrestrial proficiency of early pterosaurs

The considerations of early pterosaur limb and limb girdle functions offered here suggest views of non-pterodactyloid palaeobiology may warrant more detailed assessment. Existing models of pterosaur locomotory mechanics, where pterodactyloids are adaptable, 'terrestrialised' pterosaurs and their forebears were confined to climbing and flying, are perhaps over-simplistic. Not only are common arguments for terrestrially-inept early pterosaurs problematic, but anatomies consistent with fully erect stances and other possible hallmarks of competent terrestriality seem to be deeply nested within Pterosauria. These findings are the latest in a series showing that pterosaur palaeobiology is much richer, more diverse and complex than previously anticipated (see *Witton, 2013* for an overview).

Assessing the evolutionary pathways of the anatomies described here is complicated by the lack of consensus over non-pterodactyloid phylogeny (*Unwin, 2003*; *Kellner, 2003*; *Wang et al., 2009*; *Kellner, 2010*; *Dalla Vecchia, 2009*; *Lü et al., 2010*; *Lü et al., 2012*; *Witton, 2013*; *Andres & Myers, 2012*; *Andres, Clark & Xu, 2014*). Some tentative conclusions may be drawn, however. The distribution of glenoid and humeral morphologies identified above is complicated, with no set of features limited to specific clades or 'grades' of pterosaurs (Fig. 7). Potential signatures of erect postures appear early in pterosaur evolution: *Dimorphodon* indicates that symmetric glenoids and pterodactyloid-like humeral features had developed by the Sinemurian at the latest, and pterosaurs with elongate, robust limbs (e.g., *Peteinosaurus*, "*Eudimorphodon*" specimen MCSNB 3359, *Preondactylus*) represent some of the oldest known pterosaurs (Carnian/Norian). Given that likely pterosaur outgroups such as dinosauromorphs and *Scleromochlus* bore strong, erect limbs (e.g., *Sereno, 1991*; *Benton, 1999*), it is possible that these early pterosaurs retained characteristics of efficient terrestriality from immediate pterosaur ancestors. This might be in keeping with models of pterosaurs evolving from terrestrially- or scansorially-adapted ancestors

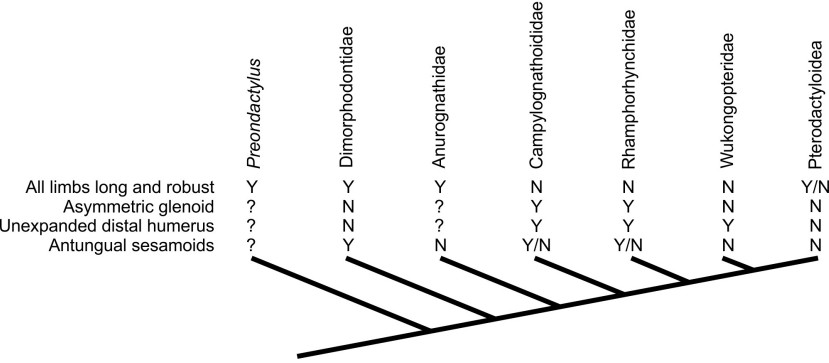

**Figure 7 Distribution of characteristics linked to terrestrial capabilities in non-pterodactyloids in a simplified pterosaur phylogeny (based on _Lü et al., 2012_).** Other pterosaur phylogenies suggest different topologies of non-pterodactyloid taxa (see text for details), but the distribution of these characteristics would be as complex, if not more so, in competing arrangements.

in inland environments (_Padian, 1985_; _Padian, 2008a_; _Bennett, 1997b_; _Witton, 2013_; _Andres, Clark & Xu, 2014_), before spreading to marine habitats (_Andres, Clark & Xu, 2014_). Taxa likely utilising sprawling forelimbs tend to occur further from the pterosaur root however, suggesting this 'traditional' stance might be a derived feature of clades such as Rhamphorhynchinae and Campylognathoididae, and perhaps associated with the development of increasingly pelagic lifestyles (see below).

How the development of these features relates to possible evidence for fully erect limbs in the pterodactyloid sister group, Wukongopteridae, is intriguing. Their Callovian/Oxfordian (_Lü et al., 2010_) appearance in the fossil record approximates the appearance of pterosaur footprints as well as the first pterodactyloids (_Andres, Clark & Xu, 2014_), making questions about distinguishing pterodactyloid tracks from those of non-pterodactyloids all the more pertinent (_Lockley, Harris & Mitchell, 2008_). Were wukongopterids involved in an upper Jurassic 'terrestrial radiation' of pterosaurs, and was this part of a separate 'terrestrialisation event' to that potentially indicated by earlier, _Dimorphodon_-like pterosaurs? Do wukongopterids represent a lineage of pterosaurs which retained plesiomorphic glenoid and humeral morphologies from much earlier pterosaurs, or were these reversed from sprawling ancestors? Future discoveries of Jurassic and Triassic pterosaurs in terrestrial basins and further resolution on the phylogeny of early pterosaurs may shed light on these questions.

Concerning the specifics of terrestrial locomotion in different non-pterodactyloid taxa: the view of early pterosaurs as forelimb-sprawling terrestrial locomotors (e.g., _Wellnhofer, 1975_; _Padian, 2008b_) seems appropriate for at least rhamphorhynchines and campylognathoidids (Fig. 8A), although how limiting their sprawled or crouched forelimbs were to walking and running remains to be determined. Padian's (_Padian, 1983b_; _Padian, 2008a_; _Padian, 2008b_) suggestion that the torsos of quadrupedal pterosaurs with sprawling forelimbs would be anteriorly inclined, and thus ill-suited to terrestrial locomotion, is questionable. As demonstrated by the alternative reconstructions of such pterosaurs provided in Fig. 5, torso inclination seems reliant on assumptions made when

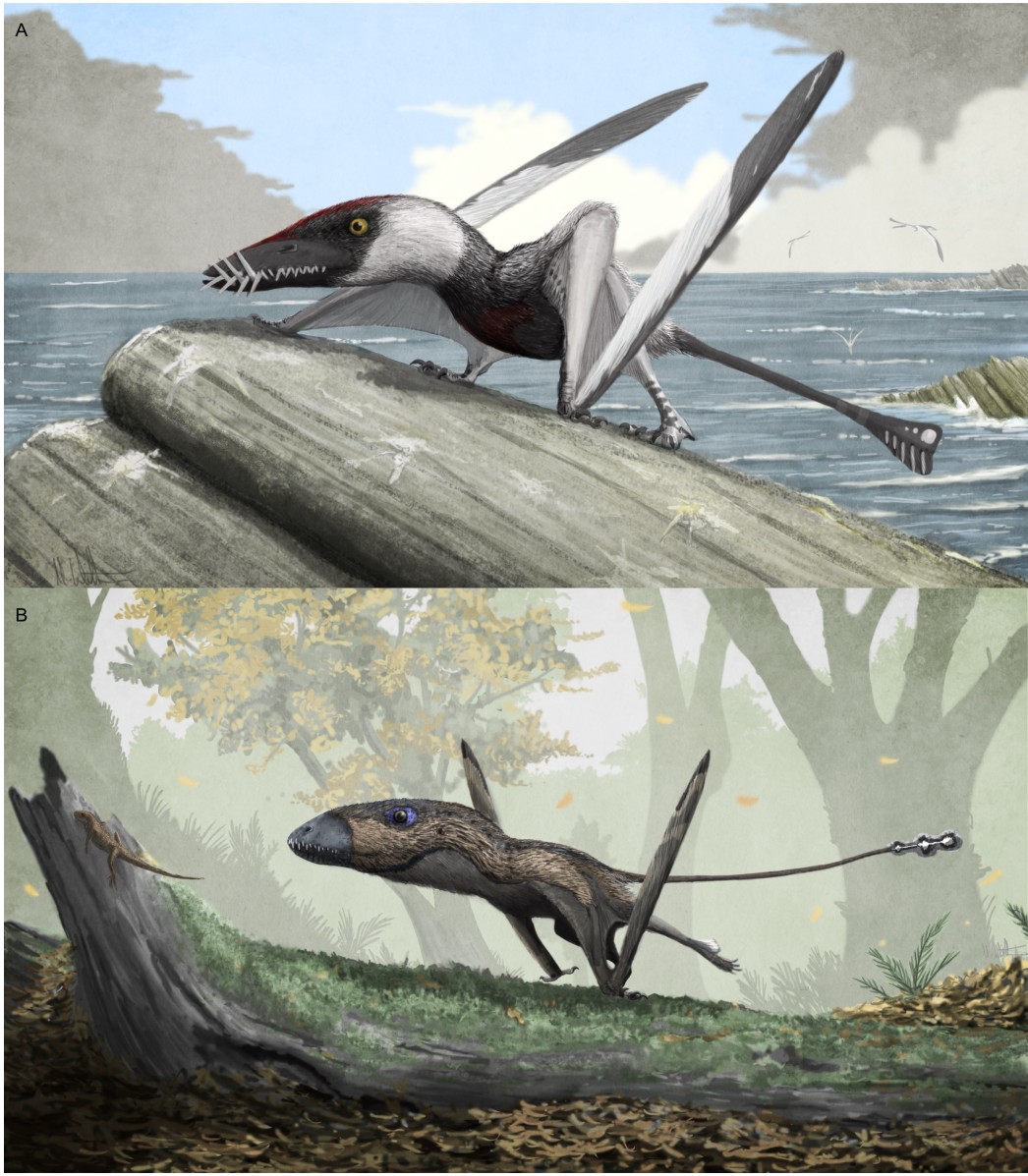

**Figure 8** **Potential variation in terrestrial locomotion gait in non-pterodactyloid pterosaurs.** (A) life restoration of the Early Jurassic rhamphorhynchine *Dorygnathus banthensis* with obligated crouching, somewhat sprawled forelimbs; (B) life restoration of the Early Jurassic species *Dimorphodon macronyx* with fully adducted humeri and parasagittal gait, shown here facilitating subcursorial, rapid terrestrial locomotion in pursuit of sphenodontian prey. Both animals are restored with retracted claws on digits possessing antungual sesamoids.

restoring pterosaur skeletons—such as enhancement of hindlimb height through elevated (digitigrade) ankles (compare Figs. 5E and 5F with *Padian*'s *2008b Dorygnathus* illustration in Fig. 1C). In any case, the fact that numerous fossil and extant quadrupedal animals have anteriorly-sloping backs and variable limb girdle heights when standing and walking (examples include protorosaurs, modern and fossil crocodylomorphs, several dinosaur

clades (diplodocoids, stegosaurids, ceratopsids), and many small mammals: lagomorphs, rodents, certain bats) questions what significance this observation has on terrestriality. Other views that sprawling gaits are inherently 'primitive' or inferior to erect ones, or somehow limit movement speed (*Unwin, 2005*) are problematic, as demonstrated by the tremendous success of sprawling tetrapods both today and in Deep Time (*Russell & Bels, 2001*). Although perhaps ill-suited to sustained terrestrial locomotion, sprawling can be an effective, perhaps superior locomotory kinematic for rapid acceleration, sprinting and climbing (*Russell & Bels, 2001*). Indeed, specialist lifestyles promote the retention or development of sprawling limbs in many species (*McElroy, Hickey & Reilly, 2008*). Thus, the sprawling forelimbs of rhamphorhynchines and campylognathoidids are not necessarily means to assume low terrestrial competency, and arguments that sprawling pterosaurs would be limited to slow, ponderous locomotion do not reflect the sometimes explosive and powerful abilities of modern sprawling amphibians, reptiles and mammals (contra. *Unwin, 2005*; *Padian, 2008b*). Note that the limbs of galloping vampire bats are sprawled (*Riskin et al., 2006*), a fact worth considering when arguing that rhamphorhynchines and campylognathoidids required bipedal stances for rapid terrestrial movement (e.g., *Padian, 2008b*; *Padian, 2008c*).

Nevertheless, because sustained terrestrial locomotion seems generally better served by erect limbs, the indication that rhamphorhychines and campylognathoidids had sprawling forelimbs might be consistent with predictions that these pterosaurs were relatively flight-reliant, seabird-like species (see functional and palaeoecological evidence discussed by *Wellnhofer, 1975*; *Wild, 1978*; *Chatterjee & Templin, 2004*; *Witton, 2008*; *Witton, 2013*, etc.). Like some seabirds, these pterosaurs may have relied on flight for long-distance movement rather than terrestrial locomotion, and their anatomy may reflect adaptive biases towards the former (e.g., *Kaiser, 2007*; *Abourachid & Höfling, 2012*). For instance, parallels may be drawn between the restricted shoulder arthrology of asymmetric glenoids and the energy-saving arthrological 'locks' found in the shoulders of modern soaring birds (e.g., *Meyers & Stakebake, 2005*). If antungual sesamoids are, as proposed here, indicators of routine claw interaction with the ground, their presence in *Dorygnathus*, *Carniadactylus* and *Eudimorphodon* still suggest frequent terrestrial activities however. It may be that these pterosaurs routinely landed to forage or roost but performed only limited walking or running activities when grounded, while other habits—perhaps hanging or climbing— necessitated large, trenchant claws and associated sesamoids. The sprawling stance of their forelimbs is well suited to climbing behaviour (*Russell & Bels, 2001*), as are the particularly large and robust third manual digits of *Dorygnathus* (Fig. 6C; *Padian, 2008b*).

With symmetrical glenoids and pterodactyloid-like distal humeri, it is possible *Dimorphodon* and wukongopterids could utilise fully upright gaits and had pterodactyloid-like terrestrial capabilities. *Dimorphodon* particularly embodies many 'subcursorial' features (long, robust limbs; stout propodia, reduced fibulae, etc.) and it may have been capable of not only sustained, but also relatively fast terrestrial activity (Fig. 8B). Especially well-developed appendages and possession of antungual sesamoids on all clawed digits might signify that *Dimorphodon* was not unduly reliant on flight, as do data suggesting it

was particularly heavy for its wingspan and a relatively ineffective, perhaps short-distance flier (*Brower & Veinus, 1981*; *Witton, 2008*; *Witton, 2013*; *Henderson, 2010*). Scansorial features of the *Dimorphodon* skeleton (e.g., elongate penultimate phalanges, asymmetrical pes structure, claw curvature—see *Unwin, 1988*; *Clark et al., 1998*; *Witton, 2013*) marry with conclusions drawn here to present it as a terrestrial generalist, capable of running, walking and climbing as well as flight. Many extant terrestrial animals with mobile limb joints and long limbs—such as rodents, bovids, carnivorans, etc.—are as adept at climbing as they are walking and running, to the extent that some measures of their ecomorphospace overlap significantly with scansorial animals (e.g., *Samuels, Meachen & Sakai, 2013*): the identification of climbing adaptations in some early pterosaurs does not preclude terrestrial proficiency. The suggested diet of insects and small vertebrates for *Dimorphodon*, based on its skull morphology, tooth shape and dental wear patterns (*Ősi, 2011*), is concordant with generally terrestrial habits (Fig. 8B). Wukongopterid pterosaurs may have also been capable terrestrial locomotors, although their limbs are not as powerfully built as those of *Dimorphodon* and some aspects of their flight anatomy, such as their pteroids, are more substantially developed (*Witton, 2013*). Like many small modern birds, wukongopterids may have been proficient enough to move through terrestrial settings without flight—perhaps in search of insect prey (*Lü et al., 2011*; *Witton, 2013*)—but they seem more aerially capable and flight-ready than the heavyset *Dimorphodon*.

## CONCLUDING REMARKS

The assessment of non-pterodactyloid terrestrial locomotion offered here demonstrates that our understanding of functionality and locomotory mechanics in early pterosaurs is limited to the extent that basic differences in limb skeleton construction have yet to be appreciated in detail. It is hoped this work will inspire further investigation into the functionality of these animals. As here, such studies will likely be hampered by the quality of non-pterodactyloid fossils, where even complete specimens can be too extensively crushed or poorly preserved to show the anatomies needed for functional interpretation. However, there is clearly greater potential for understanding early pterosaur functionality than currently realised and, until this has been researched more thoroughly, caution is urged against making generalisations about the terrestrial competency of non-pterodactyloids, and its role in the evolution of Pterosauria.

**Institutional Abbreviations**

| | |
|---|---|
| **BSP** | Bayerische Staatssammlung für Paläontologie und Geologie, Berlin, Germany |
| **HGM** | Henan Geological Museum, Zhengzhou, China |
| **GPIT** | Geologisch-Paläontologisches Institut und Museum, Universität Tübingen, Germany |
| **GSM** | Geological Survey Museum, Keyworth; UK |
| **IVPP** | Institute of Vertebrate Palaeontology and Palaeoanthropology, Beijing, China |
| **JPM** | Jehol Paleontological Museum, Chengde, China |
| **MCSNB** | Museum Civico di Scienze Naturali di Bergamo, Italy |

| MFSN | Museo Friulano di Storia Naturale, Udine, Italy |
| MJML | Museum of Jurassic Marine Life, Kimmeridge, UK |
| MPUM | Museo Paleontologia Universitá di Milano, Italy |
| NHMUK | Natural History Museum, London, UK |
| PIN | Paleontological Institute, Moscow, Russia |
| SMNS | Staatliches Museum für Naturkunde Stuttgart, Germany |

## ACKNOWLEDGEMENTS

Lorna Steel and Sandra Chapman are thanked for repeated access to specimens in their care and for assistance obtaining literature. Michael Habib, Luis Chiappe and an anonymous referee are thanked for providing comments on this work, as are Hans Thewissen, Peter Binfield and Matt Wedel for their editorial contributions. David Martill, Liz Martin and Michael O'Sullivan are thanked for discussions of early pterosaur functionality. David Hone, Ross Elgin, David Unwin and Calum Davies are thanked for sharing excellent photographs of German and Chinese pterosaur specimens.

### Funding

The author declares there was no funding for this work.

### Competing Interests

The author declares there are no competing interests.

### Author Contributions

- Mark P. Witton conceived and designed the experiments, performed the experiments, analyzed the data, contributed reagents/materials/analysis tools, wrote the paper, prepared figures and/or tables, reviewed drafts of the paper.

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
