# Peer review of "Were early pterosaurs inept terrestrial locomotors?"

_PeerJ, doi:10.7717/peerj.1018_

## Round 0.1 · original submission · Major Revisions

This paper reads more like an essay than a scientific research paper. Most of the original research was limited to two pterosaurs in British museums, while other specimens have not been studied even though sweeping generalizations are made. Furthermore, there was a lively discussion of bipedality in pterosaurs a few decades ago. Some of that work is cited, but some is not and much of it is treated in a rather cursory fashion.

This manuscript should be cut down to the areas where there is original research. The section on the uropatagium has none, the most original section in it is based on anecdotal literature descriptions of how some modern vertebrates with uropatagia move. The section on trackways appears to just be literature review. These two sections should be reduced in length and may be appropriate as short paragraphs in the discussion of the manuscript. The discussion about sprawling gait again does not evaluate past work critically, but this section does contain original observations and a discussion of the glenoid. This section could be expanded. Morphologies should be better documented and this would form the core of this manuscript. The last section, on antungual phalanges, again is mainly a literature review and, given that its conclusion is weak, appears to not contribute much to the overall conclusion. This section too, should be shortened.

·

Basic reporting

No Comments

Experimental design

No Comments

Validity of the findings

No Comments

Comments for the author

Page 5: “This is thought to reflect the general lack of inland or near-shore Lagerstätten in the Triassic and Jurassic”

I suspect this is meant to say “Early-Middle Jurassic” (i.e. before the Late Jurassic known late-Jurassic Lagerstätten such as the Solnhofen).


Page 8: “These taxa are characterised by erect limbs, the preacetabular process anchoring large hindlimb extensors for moving the hindlimb forward in the parasagittal plane (Hyder et al. 2014).”

That should make the preacetabular muscles hip flexors, rather than extensors. I suspect that “hindlimb extensors” is used here in reference to the extension plane for the rest of the hindlimb, but I suggest that it will be more clear to refer to the specific joint(s) and their motions, rather than the limb as a whole. (i.e changing “hindlimb extensors” to “hip flexors”).


Page 10: “Other than Dimorphodon, only the wukongopterids Darwinopterus linglongtaensis (IVPP V16049, Wang et al. 2010) and Darwinopterus robustodens (HGM 41HIIII-0309A; Lü et al. 2011) seem to possess symmetric glenoids”

It might be useful to clarify indicating specifically that these glenoid categories do not apply to pterodactyloids before this section, or by adding “among non-pterodactyloid pterosaurs” to the end of the above sentence. It becomes apparent just after this section, but at first it seems odd because I was thinking that many pterodactyoid glenoids more or less fit with the “symmetric” category.


Page 11: “campylognathoidids” does not require the plural “s”


Page 14: “… in any case, climbing or hanging would strain the ventrally-situated digit flexors, not the extensors, contradicting most interpretations of sesamoid function (e.g. Haines 1969)”

This is not necessarily true, though the point is well taken. There flexor tendons would experience significant negative strain (i.e. becoming shorter), but the extensor tendons undergo significant positive (tensile) strain during grasping, and they can be important in maintaining grip. Since the flexors are the primary source of grasping power, the *stress* will be much greater on the flexor side, but the strain difference may not be as large. Rapid opening (i.e. extension) of the manus can also be important in some climbing taxa (e.g. geckos). I note that claw extension is covered at the end of this page and continuing on to page 15, so that may not require any additional considering (though the observation of rapid hyperextension of the claws in geckos does work in contradiction to the Padian and Olsen 1984b commentary that modern reptiles do not retract their claws).

·

Basic reporting

This is a very interesting paper, well-written and arriving at very reasonable conclusions. In all, I don't have any comments regarding the manuscript. The conclusions are sound and the author has made a good case defending his arguments.

I do have suggestions regarding the illustrations:

1. The quality of the reproduction of Dimorphodon macronyx on Figure 1A is very poor. The author should try to either scan it again (and better) or retrace it.

2. It would be helpful to have a general illustration indicating the location of the various patagia in pterosaurs. This would help the general reader.

3. The light blue silhouettes on Figure 6 are rather useless. They should either be better or they should be removed.

4. I don't find Figure 7 to be very useful. I don't really think it is necessary.

Experimental design

No comments

Validity of the findings

See my comments above.

Comments for the author

See my comments above.

·

Basic reporting

Review communicated privately

Experimental design

Review communicated privately

Validity of the findings

Review communicated privately

Comments for the author

Review communicated privately

---

## Round 0.2 · Minor Revisions

I was asked to adjudicate the fate of this manuscript following a decision by another Editor to reject it, and an Appeal of that decision by the author. I will briefly explain the reasoning behind my decision, both so that it can be better understood by everyone involved in the current case, and hopefully also to inform future decisions (whether or not the editors making those decisions agree with my arguments).

The rejection decision is based on three points, which I will address in turn: one of the reviewers recommended rejection; the manuscript reads more like an essay than scientific paper; and it includes more review material than new material.

1. One of the reviewers recommended rejection. That is true, but the other two did not, and compared to their relatively mild critiques, the recommendation to reject is the outlier here. Given PeerJ's aims and scope, we should err on the side of acceptance unless the case against the submission is extremely clear and compelling. First because rejections waste everyone's time. Second because it is practically impossible to tell in advance how a paper will fare post-publication - papers released with fanfare may turn out to be fatally flawed, and other works published quietly in small journals may become citation classics. This happens because the worth of a paper is ultimately judged by its readers, and we should have some faith in that process. Third because there are already mechanisms in place for de facto post-publication review – if any reviewers or readers are unhappy with the publication of this manuscript, they can leave comments on the published paper or write rebuttals. Beyond these points, I cannot find the case for rejection very compelling when two out of three qualified reviewers recommended acceptance.

2. The manuscript reads more like an essay than a scientific research paper. I must confess, I don't know what that criticism means, or why it amounts to more than a tone argument. Either the reasoning is solid or it isn't. I read both versions of the manuscript and found the prose highly readable - surely this is a feature and not a bug. Historically and etymologically, an essay is literally 'an attempt' - an attempt to understand something by working through its underpinnings and implications. There is nothing inherently unscientific about that, and some of the most important scientific works of the past few centuries were written informally so that they could be understood by a broad audience.

3. The manuscript is more of a review than an original contribution. We're at a strange time in paleontology when we're finding new ways to apply existing knowledge from other fields to paleobiological problems. One of the side-effects of this is that some papers are hard to classify. If an author uses only pre-existing data but applies it to a novel problem to yield a novel result, is that review paper or not? I would say no – we should distinguish between true reviews, which only summarize existing knowledge, and synthesis papers, which may draw from a spectrum of pre-existing and novel data to present a novel conclusion, or at least weigh in on an outstanding controversy. If we make that distinction, this manuscript is clearly a synthesis piece, not a review paper. Also, following the first point above, we should err in favor of publishing papers that make novel contributions. Disqualifying manuscripts because they rely in part on pre-existing data strikes me as both legalistic and ungenerous.

Given all of the above, and after carefully reading both versions of the manuscript, the reviewers' comments, and all of the editorial correspondence, I have decided to accept the manuscript pending minor revisions. In revising the manuscript for publication, I urge you to consider the following points:

- In his review, Kevin Padian contended that this is not a scientific paper because you are not testing any hypotheses. Based on my reading, the hypothesis that basal pterosaurs were poor terrestrial locomotors is widely accepted, and in this manuscript you are addressing the solidity of that hypothesis by reference to taphonomy, functional morphology, analogies with extant animals, and so on, which is as much of a test as many paleobiological hypotheses ever get. You might want to make that more explicit.

- I found a couple of places in the revised manuscript where terms related to stance and gait are still used incorrectly, e.g., "erect gait" instead of "parasagittal gait". These and other related terms were recently clarified by Padian et al. (2010, Palaeontology), which you should refer to and cite. I also recommend doing a global search of the manuscript to make sure that terms related to posture and locomotion are used correctly.

- Finally, assuming that you are willing to make the requested revisions and that the manuscript is published in due course, it will probably be contentious, at least based on Kevin Padian's critique. Indeed, even if you elect not to publish the peer review history, there is nothing to stop Kevin from publishing the review elsewhere, either as-is or in the form of a rebuttal paper. With that possibility (or perhaps inevitability) in mind, I urge you to do one more pass on the manuscript, just to make sure that you've been as generous as possible to previous researchers, and that you have represented their evidence and arguments accurately and fairly.

· Appeal

Appeal

Dear PeerJ editors,

Thank you for considering this appeal. I want to apologise for taking up your time with this, and stress that this is the first time I have made an appeal against an editorial decision at any journal. However, I consider my recent (and to date, only) PeerJ editorial experience to warrant a second opinion, or at least more clarification than my editor (J. Thewissen) was able or willing to provide.

Following review and re-submission, my solo-authored manuscript “Were early pterosaurs inept terrestrial locomotors?” has been rejected from PeerJ on grounds that it is more of an “essay-like review” than a research paper, and thus unsuitable for PeerJ. I disagree with this for a number of reasons and consider several aspects of the MS editorship questionable. In all, my MS rejection from PeerJ has left me feeling misled on your journal scope and fear that the time of myself and three others – my referees – have been wasted in pursuit of its publication with you. I hope this can be rectified – or at least clarified - via this appeal, and that a decision can be reached quickly.

My objections to the decision and issues with editorial process are provided below. I trust you have access to relevant manuscript files and my rebuttal letter – my responses in the former are referenced below, so it may be worth taking a look at it alongside this. Please let me know if I can be of any assistance in this process.

Thanks for your time,

Mark Witton

1. MS style, data presentation and originality, and suitability for PeerJ

My MS has been identified as reading like an ‘essay-like review’ by the editor in both rounds of comments, as well as in correspondence (referee 3 provided comments as a separate file, leading to some email exchange between myself and the editor as they were exchanged. I can provide these if required). I strongly dispute that my MS is little more than a review of existing literature. I certainly discuss and review literature, but only that which provides insight on the paper topic - questioning the current status quo on non-pterodactyloid pterosaur terrestrial locomotion. For instance, I discuss the widely-cited hypothesis that hindlimb-spanning membranes limited the terrestrial capabilities of pterosaurs. In doing so, I mention documented accounts of proficient terrestriality in modern animals with pterosaur-like membranes because they obviously challenge this interpretation: such use of analogy is common in papers on functional morphology in extinct animals. Such an argument has not been raised in pterosaur literature before, necessitating a short review of relevant literature – no more than a few hundred words – to provide context before it can be compared to what we know of pterosaurs and their relevant anatomies. I see this as a means of bringing relevant data, new to the debate on pterosaur locomotion, to challenge an existing hypothesis. Another section, questioning claims that the stratigraphic occurrence of pterosaur tracks is 'significant' with respect to pterosaur terrestriality is similarly constructed, highlighting how perceptions of pterosaur trackways are unusual against ichnological perceptions for other extinct animals, among other points. Some components of the paper are not reliant on review at all, instead presenting new data concerning the shoulder and forelimb anatomy of early pterosaurs, and discussing its significance with respect to forelimb stance. The MS covers more ground than this, and contrary to the editor, I argue that there is a lot of new information and insight here. The suggestion in my rejection letter that the former two sections are longer than the description and interpretation of pterosaur forelimbs is incorrect: presentation of discussions of shoulder and forelimb anatomy

are at least 60% longer (c. 1000 words each for the former, and c. 1600 for the later (not including subsequent discussions of forelimb posture in the discussion section)).

In his formal response to my first submission, the editor stated that "Reading the reviews confirmed the editors’ opinion that much of your manuscript is a review of previous work". This is also in error. Two of three referees found no significant issues with the paper, and one even described it as “a very interesting paper, well-written and arriving at very reasonable conclusions.” Neither mentions anything about my MS being a review or presenting no new information. Referee 3 clearly thinks I said nothing new, but he also chronically mischaracterises and misinterprets MS content on multiple occasions, as outlined at length in my rebuttal letter. This response to referee 3, which undermines most of his comments, has not been cause for reconsideration about the reviews ‘confirming’ the editors opinion.

An important component of my appeal is that PeerJ has already published papers written in a similar style to my own submission. I am thinking specifically of Taylor and Wedel’s (2013) assessment of sauropod neck length (https://peerj.com/articles/36/), a piece which inspired my choice of PeerJ as a publication venue as well as the style of my article. Being aware that PeerJ does not publish straight review articles, I was careful to check the review history of this paper to see how it had been handled. The Taylor and Wedel piece had been identified by the editor (J. Hutchinson) as ‘review-like’ but with sufficient novel insight to warrant publication (https://peerj.com/articles/36/reviews/). A similar mix of review and novel insights (leaning towards the latter, including new fossil descriptions and comparisons) was the goal of my MS. The review history of Taylor and Wedel (2013) (along with the fact that one these authors is now PeerJ editor (Matt Wedel) and presumably knows what a PeerJ paper should be like), suggested to me that this 2013 paper was a suitable model for prospective PeerJ papers, including my own. The fact my MS has been rejected primarily for following an existing PeerJ style seems quite unreasonable.

When approaching Hans with the above point about previous PeerJ content, he has only responded stating he “cannot speak for the other associate editors and the decisions they made about other manuscripts". Although I appreciate editorial decisions are somewhat subjective, it seems that there is clear inconsistency in editorial perception here. If the styles of past papers and their published review comments cannot be trusted for guidance, how are authors meant to reliably gauge journal remit? For authors, published content is a much better indicator of accepted styles than policy statements, which are clearly – as demonstrated by your own editorial board – open to interpretation.

2. Editorial decisions leading to wasted time and complicating manuscript changes

With the above points in mind, I hope you can understand my frustration concerning the fact that the MS was sent out for review before major editorial issues were raised. In our email correspondence following peer review, Hans suggested my MS was always “problematic in its scope” with respect to its suitability for PeerJ, and that his views were allegedly “confirmed” by the referees. Why was such a 'problematic' MS sent for peer-review? It seems obvious that these editorial changes should have been discussed before the MS was accepted for review, where I could have either pulled the MS or decided to reconfigure it accordingly. Instead, the time of four individuals - myself and the three referees – have potentially been wasted in the review process of an MS the editor considered ill-suited to the journal from the start.

This decision has complicated my revisions to the MS. Two of my referees consider the MS in good shape: they only recommend very minor changes to wording, content and illustrations. Referee 3 provides a very negative account of the paper, but I feel I have demonstrated his review constantly misunderstands the MS and relies on an idiosyncratic view of pterosaur palaeobiology, to the extent that most of his comments are of little value. The controversial views of this individual are well known among the pterosaur community. However, the editor leaned heavily on this review in his formal response to my original submission, apparently seemingly unaware of some current consensuses on the animals discussed in the paper (e.g. the suggestion the MS needs greater discussion of pterosaur bipedality, a widely unfavourable perception of pterosaur locomotion for all but referee 3 – please see my rebuttal letter for more on this). My response to referee 3 has not been commented on by the editor, so I am unaware of his current take on the scientific content of my MS – the rejection seems largely based on my failure to reduce the alleged ‘review’ components of the MS.

Controversy of referee 3 aside, I have an MS that, content wise, at least two pterosaur experts have suggested was virtually ready for publication. As outlined in my response letter, I have made all changes I felt reasonable by the referees and editor, including modifying figures, making the MS more concise where possible and providing greater explanation/clarity where needed. However, additional editorial changes involve considerable removal of material identified as “well-written”, “interesting”, and “well made” by two referees, and expansion of areas they thought were already clear and well-argued. I have attempted expanded these sections somewhat while keeping the original format of the paper, but these changes are not viewed as sufficient, the editor stating the MS needs even more expansion (even some – pertaining to antungual sesamoid function – which he asked to be shortened in his first round of comments). I find myself being rejected even though two of my peers have given the MS good reviews, and instead because I did not modify the MS from a style already represented at PeerJ. I do not understand this decision.

3. Lack of editorial discourse

Finally, it seems worth stressing that this appeal is partly due to a lack of discussion from the editor on the points raised here. I have outlined much of the above to the editor in both email correspondence and ‘formally’ in my rebuttal letter, but have received little response to them. Statements like that quoted above (“I cannot speak for the other associate editors and the decisions they made about other manuscripts”) and “I have read the response to my request for change in this regard, but do not find it compelling” are about the limit of discussion. It seems that any query on editorial decisions is not considered, despite my requests being reasonable. Is it out of the question to expect an editor to check another paper in his journal, or ask colleague about a decision, following a direct line of enquiry on existing journal style? And especially so in a journal where the reviewing process is available for all to see? This lack of discourse has been very unhelpful, and contrasts markedly with what my expectations from a journal lauded for being progressive, open and helpful to authors. Greater communication between myself and the editor in this case – both post-review and before – may have avoided this unpleasantness altogether.


· · Academic Editor

Reject

This manuscript has improved over the previous version, and many of the minor problems have been corrected. However, the manuscript still reads like an essay and is more review than new material. For instance, the length of the first and second question in the results section is longer than the new material on the shoulder in the third question. I have read the response to my request for change in this regard, but do not find it compelling. From his comments, it is clear that the author has no interest in adjusting the problems not dealt with, so I recommend rejection.

There is material in this manuscript that can be salvaged, improved descriptions with explicit analyses on the shoulder joint and antungual phalanges would be interesting. But that would mean basically writing a new manuscript.

As it stands, I do not consider this manuscript publishable in PeerJ.

---

## Round 0.3 · accepted · Accept

Thank you for the swift and thorough revision. I am happy to accept your paper for publication in PeerJ.